# CLASS-CONDITIONAL DOMAIN ALIGNMENT VIA KERNEL CAUCHY-SCHWARZ MUTUAL INFORMATION

## ABSTRACT

Domain Generalization (DG) seeks to learn models that are robust to unseen distribution shifts, a critical challenge for real-world machine learning applications. A dominant paradigm is to enforce domain invariance by aligning feature distributions from multiple source domains. However, aligning marginal feature distributions indiscriminately can discard critical class-discriminative information, especially when class priors vary across domains. We address this limitation with Domain Alignment via Kernel Cauchy-Schwarz Mutual Information (DAS-MI), a novel framework that advances principled class-conditional alignment. The core principle is to maximize the statistical dependence between same-class features across different domains. We operationalize this using the Cauchy-Schwarz Quadratic Mutual Information (CS-QMI), a powerful information-theoretic measure. Critically, and in contrast to prior work relying on complex approximations or adversarial training, our approach yields a closed-form, non-parametric alignment objective derived from kernel density estimates. This results in a stable loss that integrates seamlessly into deep learning pipelines. Extensive experiments across five benchmark datasets demonstrate performance comparable to state-of-the-art methods. DAS-MI offers a theoretically-grounded and practically efficient solution to domain generalization that robustly preserves discriminative information.

## 1 INTRODUCTION

A foundational assumption underlying most machine learning algorithms is that the training and test data are drawn from the same distribution, a condition known as the independent and identically distributed (i.i.d.) assumption (Zhou et al., 2023). In practice, however, this assumption often does not hold due to distribution shifts arising from variations in data collection processes, environmental conditions, or demographic differences, leading to significant performance degradation when models are deployed in real-world, out-of-distribution (OOD) scenarios. This highlights the need for models with the property of Domain Generalization (DG), i.e., models that generalize robustly across unseen domains without access to samples from these domains during training.

To address the challenge of domain shift, Domain Generalization (DG) was formally introduced in (Blanchard et al., 2011). In the DG setting, a predictive model is trained using data from multiple source domains and subsequently evaluated on an unseen target domain. A growing body of literature has identified feature invariance as a central principle for improving OOD generalization. Foundational theoretical results (Ben-David et al., 2010) have established explicit bounds that relate the generalization error on unseen domains to the divergence between the corresponding feature distributions. In particular, it has been demonstrated that reducing the discrepancy among domain-specific feature distributions across domains constitutes a principled mechanism for mitigating risk on unseen domains (Muandet et al., 2013). Motivated by this theoretical foundation, a variety of approaches have been proposed to achieve distributional alignment, employing strategies such as Maximum Mean Discrepancy (MMD) minimization (Li et al., 2018c), adversarial feature alignment, and domain-specific regularization techniques (Sun & Saenko, 2016). Nevertheless, relying solely on moment-matching or adversarial alignment can be insufficient. Moment-based methods often capture only lower-order statistics (e.g., means and covariances), potentially failing to align complex, non-linear distributional shifts (Wang et al., 2023a). Similarly, adversarial approaches, while powerful, involve unstable min-max optimization games that essentially approximate distri-

bution matching. To rigorously enforce domain invariance, we require a fundamental measure of statistical dependence that captures all orders of interaction without the instability of adversarial training.

Mutual information (MI), an information-theoretic measure of statistical dependence, has been used in recent work on DG in order to operationalize invariance, e.g., by reducing the dependence between learned features and domain identity (Nguyen et al., 2021; Dong et al., 2025). However, aligning marginal feature distributions indiscriminately can discard critical class-discriminative information, especially when class priors vary across domains (label shift). In such scenarios, minimizing marginal MI forces the alignment of features from different classes, leading to negative transfer. We address this limitation with Domain Alignment via Kernel Cauchy-Schwarz Mutual Information (DAS-MI), a novel framework that advances principled class-conditional alignment. The core principle is to maximize the statistical dependence between same-class features across different domains. We operationalize this using the Cauchy-Schwarz Quadratic Mutual Information (CS-QMI) (Yu et al., 2024b), a powerful information-theoretic measure. While CS-QMI has been recently employed for information bottleneck in regression (Yu et al., 2024b), we extend its utility to Domain Generalization by formulating a class-conditional estimator that maximizes cross-domain dependence. Critically, and in contrast to prior work relying on variational approximations or adversarial networks, our approach yields a closed-form, non-parametric alignment objective derived from kernel density estimates. This results in a stable loss that integrates seamlessly into deep learning pipelines. Extensive experiments across five benchmark datasets demonstrate performance comparable to state-of-the-art methods. DAS-MI offers a theoretically-grounded and practically efficient solution to domain generalization that robustly preserves discriminative information.

To summarize, we make the following contributions:

- We formalize a link between class-conditional invariance and risk minimization by connecting class-conditional domain generalization bounds to a kernelized Cauchy-Schwarz discrepancy minimized through maximizing class-conditional CS-QMI across domains.

- We introduce a scalable, nonparametric estimator of class-conditional CS-QMI based on kernel Gram matrices, enabling a simple, closed-form alignment loss that is numerically stable and easy to integrate into deep networks.

- We demonstrate performance that is comparable with (and sometimes outperforming) the state-of-the-art on standard DG benchmarks.

In summary, DAS-MI bridges generalization theory and practical learning. By maximizing class-conditional CS-QMI across domains, it enforces the invariance structure most relevant for DG while remaining computationally efficient and broadly applicable.

## 2 THEORETICAL FOUNDATION: INVARIANCE AND RISK IN DOMAIN GENERALIZATION

### 2.1 PROBLEM SETUP AND OBJECTIVE

Let $\mathcal{E}_{\mathrm{s}}$ be a set of observable source domains and $\mathcal{E}_{\mathrm{all}}$ be the superset containing all possible domains, including unseen target domains. Each domain $e \in \mathcal{E}_{\mathrm{all}}$ induces a joint probability distribution $P^e$ over an input space $\mathcal{X}$ and a label space $\mathcal{Y}$. [1] For a given hypothesis (i.e., a predictive model) $h : \mathcal{X} \to \mathcal{Y}$. Let $Z = \Phi(X) \in \mathcal{R}$ be a learned representation (encoder), $h = g \circ \Phi$ such as $g$ is the classifier, and a loss function $\ell$, the population risk on a specific domain $e$ is the expected loss defined as:

$$R^e(h) = \mathbb{E}_{(x,y) \sim P^e}[\ell(h(x), y)]. \tag{1}$$

Here, $(x, y)$ is an input-label pair sampled from the distribution $P^e$. In a practical setting, we only have access to a finite dataset $D_e = \{(x_i, y_i)\}_{i=1}^{n_e}$ for each source domain $e \in \mathcal{E}_{\mathrm{s}}$. This allows us to compute the empirical risk, $\hat{R}^e(h)$, which is the average loss over the sample $D_e$. A standard

---

[1]Following standard literature in invariant representation learning, we assume that a stable labeling function exists across domains (i.e., no significant concept shift).

approach is to train a model, parameterized by $\theta$, by minimizing the average empirical risk across all source domains, i.e.:

$$\min_{\theta} \ R^s(h); \quad R^s(h) = \frac{1}{|\mathcal{E}_s|} \sum_{e \in \mathcal{E}_s} \hat{R}^e(h_{\theta}), \tag{2}$$

This approach, known as Empirical Risk Minimization (ERM), can overfit to spurious correlations specific to the source domains, leading to poor performance on a target domain $\mathcal{E}_t \in \mathcal{E}_{\text{all}} - \mathcal{E}_s$.

The goal of Domain Generalization (DG) is to overcome this limitation by learning a hypothesis that performs well on any unseen domain. This is formally expressed as a minimax problem, which seeks to find a hypothesis that minimizes the worst-case risk across all possible domains:

$$\min_{h \in \mathcal{H}} \ \sup_{e \in \mathcal{E}_{\text{all}}} \ R^e(h), \text{ where } \mathcal{H} \text{ is the space of all possible hypotheses.} \tag{3}$$

## 2.2 A High-Probability Bound for Generalization to Unseen Domains

Directly optimizing the minimax objective in equation 3 is often intractable. Instead, we ground our approach in the high-probability domain generalization (DG) framework of Dong et al. (2025), which provides a formal path to controlling risk on unseen domains.

Let $\nu$ be a meta-distribution over the space of all possible domains $\mathcal{E}_{\text{all}}$. For a given hypothesis $h$ and a domain $e \in \mathcal{E}_{\text{all}}$, the performance is measured by the risk $R^e(h)$. The framework considers three key quantities, the average risk over the meta-distribution: $R(h) := \mathbb{E}_{e \sim \nu}[R^e(h)]$, The source risk on the set of training domains $\mathcal{E}_s$: $R^s(h) := \frac{1}{|\mathcal{E}_s|} \sum_{e \in \mathcal{E}_s} R^e(h)$, and the target risk on a set of unseen domains $\mathcal{E}_t$: $R^t(h) := \frac{1}{|\mathcal{E}_t|} \sum_{e \in \mathcal{E}_t} R^e(h)$.

The core insight is to use the average risk $R(h)$ as a bridge to connect the known source risk with the unknown target risk. For any constant $\lambda \in (0, 1)$, the triangle inequality implies:

$$P\{|R^t(h) - R^s(h)| \geq \epsilon\} \ \leq \ P\{|R^t(h) - R(h)| \geq \lambda\epsilon\} \ + \ P\{|R(h) - R^s(h)| \geq (1 - \lambda)\epsilon\}. \tag{4}$$

The first term, representing the gap between the target and average risks, is the crux of domain generalization. Dong et al. (2025) provide a high-probability bound for this term. Let $D$ be a random variable for the domain index. Under a loss function bounded in $[0, M]$, they show:

$$P\{|R^t(h) - R(h)| \geq \varepsilon\} \ \leq \ \frac{M}{\varepsilon\sqrt{2}} \sqrt{I(Z; D)} + 2R^{\star}, \tag{5}$$

where $I(Z; D)$ is the mutual information between the learned representation and the domain index, and $R^{\star} = \min_g \mathbb{E}_{e \sim \nu}[R^e(g)]$ is the risk achievable by the optimal classifier.

Combining equation 4 and equation 5 via a union bound yields the final generalization gap:

$$P\{|R^t(h) - R^s(h)| \geq \epsilon\} \ \leq \ \underbrace{\frac{M}{\lambda\epsilon\sqrt{2}} \sqrt{I(Z; D)} + 2R^{\star}}_{\text{Target Deviation Term}} \ + \ \underbrace{P\{|R^s(h) - R(h)| \geq (1 - \lambda)\epsilon\}}_{\text{Source Deviation Term}}. \tag{6}$$

In words, the target-source gap is controlled by two components. The target deviation term is of primary interest, as it shrinks when the data representation becomes more domain-invariant (i.e., $I(Z; D)$ is small), directly addressing the challenge of generalizing to unseen data. The source deviation term, which measures how well the finite set of training domains represents the meta-distribution, can be bounded using standard learning-theoretic techniques. As this is not the novel focus of our work, we concentrate on the more critical target deviation term.

## 2.3 Cauchy-Schwarz Divergence and Mutual Information

Estimating Shannon MI , i.e., equation 6 reliably during training is brittle in high dimensions. We therefore move to the Cauchy-Schwarz (CS) divergence, kernel-estimable surrogate for MI between a joint and the product of its marginals.

Motivated by the Cauchy-Schwarz (CS) inequality for square integrable functions given by

$$\left( \int P(x)Q(x)dx \right)^2 \leq \int P(x)^2 dx \int Q(x)^2 dx, \tag{7}$$

with equality holding if and only if $P(x)$ and $Q(x)$ are linearly dependent, a measure of the "distance" between probability density functions (PDFs) $P(x)$ and $Q(x)$ known as the CS divergence can be defined as follows (Principe, 2010; Yu et al., 2024a):

$$D_{CS}(P(x); Q(x)) = -\log \left( \frac{\left( \int P(x)Q(x)dx \right)^2}{\int P(x)^2 dx \int Q(x)^2 dx} \right). \tag{8}$$

The CS divergence is symmetric and non-negative, with $D_{CS}(P(X); Q(x)) = 0$ if $P(x) = Q(x)$. Given i.i.d. samples $\{x_i^p\}_{i=1}^m$ from $P(x)$ and $\{x_j^q\}_{j=1}^n$ from $Q(x)$, the CS divergence can be empirically estimated in a closed form using a kernel density estimator (KDE) (Parzen, 1962) as follows:

$$\hat{D}_{CS}(\{x_i^p\}_{i=1}^m; \{x_j^q\}_{j=1}^n) = \log \left( \frac{1}{m^2} \sum_{i,j=1}^m \kappa(x_i^p, x_j^p) \right) + \log \left( \frac{1}{n^2} \sum_{i,j=1}^n \kappa(x_i^q, x_j^q) \right)$$
$$- 2\log \left( \frac{1}{mn} \sum_{i=1}^m \sum_{j=1}^n \kappa(x_i^p, x_j^q) \right), \tag{9}$$

where $\kappa$ is a kernel function, such as the Gaussian kernel $\kappa_\sigma(x, x') = \exp(-\|x - x'\|_2^2/2\sigma^2)$.

This framework can be extended to measure the statistical independence between two random variables $X$ and $Y$. By replacing the single densities $P(x)$ and $Q(x)$ with the joint distribution $P(x, y)$ and the product of marginals $P(x)P(y)$ respectively, we obtain the CS-QMI (Principe, 2010):

$$I_{CS}(X, Y) = D_{CS}(P(x, y); P(x)P(y)) = -\log \left( \frac{\left( \int P(x, y)P(x)P(y)dxdy \right)^2}{\int P(x, y)^2 dxdy \int (P(x)P(y))^2 dxdy} \right). \tag{10}$$

Unlike the standard Kullback-Leibler (KL) divergence, which is notoriously difficult to estimate, CS-QMI can be efficiently estimated from samples non-parametrically. This property allows for a robust and direct implementation of information-theoretic objectives in deep learning without requiring variational approximations or strong distributional assumptions.

## 2.4 CONNECTING THE GENERALIZATION BOUND TO A TRAINABLE OBJECTIVE

The generalization bound derived in Section 2.2 provides a clear theoretical path forward: to improve out-of-distribution performance, we must minimize the mutual information $I(Z; D)$ (equation 6). Subsequently, in Section 2.3, we introduced the CS-QMI as a robust, kernel-based tool for estimating statistical dependence. This section forges the crucial link between these concepts, showing how a practical objective based on CS-QMI directly minimizes a refined, target-relevant generalization bound.

The marginal mutual information $I(Z; D)$ penalizes any statistical difference between domain representations. However, in classification, aligning the full marginal distributions $P(Z|D = e_i)$ can be counterproductive, especially when class priors $P(Y|D = e_i)$ differ across domains. A model should instead learn representations that are invariant for a given class.

This motivates refining the bound to focus on the class-conditional mutual information, $I(Z; D|Y)$, which measures the domain information remaining in features after the class label $Y$ is known. As we prove in Appendix C, this leads to a more relevant high-probability bound on the target deviation:

$$P\left\{ \left| R^t(h) - R(h) \right| \geq \varepsilon \right\} \leq \frac{M}{\varepsilon\sqrt{2}} \sqrt{I(Z; D|Y)} + 2R^\star. \tag{11}$$

Minimizing this class-conditional term is the core theoretical principle of our method, as it directly encourages the model to learn features that are discriminative for the task ($Y$) but invariant to the domain ($D$).

Let $P_i^c := P(Z \mid Y = c, D = e_i)$ denote the class-conditional feature distribution in domain $e_i$, and let $\bar{P}^c := \frac{1}{|\mathcal{E}_s|} \sum_{i=1}^{|\mathcal{E}_s|} P_i^c$ be the corresponding per-class barycenter.

**Lemma 1** (Class-conditional mutual information as JS divergence). *Assume that, for each class $c$, the domain index $D$ is uniformly distributed over the $|\mathcal{E}_s|$ source domains given $Y = c$, and that each $P_i^c$ admits a density with respect to a common reference measure. Then*

$$I(Z; D \mid Y = c) = \frac{1}{|\mathcal{E}_s|} \sum_{i=1}^{|\mathcal{E}_s|} D_{\mathrm{KL}}\big(P_i^c \,\|\, \bar{P}^c\big) = \mathrm{JS}\big(\{P_i^c\}_{i=1}^{|\mathcal{E}_s|}\big).$$

*Consequently,*

$$I(Z; D \mid Y) = \mathbb{E}_{c \sim P(Y)}\Big[\mathrm{JS}\big(\{P_i^c\}_{i=1}^{|\mathcal{E}_s|}\big)\Big].$$

*The detailed proof is provided in Appendix C.*

Next, we make the assumptions under which the Jensen-Shannon divergence can be controlled by Cauchy-Schwarz divergences.

*Assumption* 1 (Bounded class-conditional density ratios). For each class $c$, the conditional laws $\{P_i^c\}_{i=1}^{|\mathcal{E}_s|}$ are absolutely continuous with respect to a common reference measure with densities $\{p_i^c\}$. Define the per-class barycentric density $\bar{p}^c(z) := \frac{1}{|\mathcal{E}_s|} \sum_i p_i^c(z)$ and assume that there exist constants $0 < a_c \le b_c < \infty$ such that

$$a_c \le \frac{p_i^c(z)}{\bar{p}^c(z)} \le b_c \quad \text{for all } z \text{ with } \bar{p}^c(z) > 0 \text{ and all } i \in \{1, \ldots, |\mathcal{E}_s|\}.$$

In particular, this implies that all $P_i^c$ share the same support $\{z : \bar{p}^c(z) > 0\}$ and that the density ratios $p_i^c / \bar{p}^c$ are uniformly bounded above and below on this support.

Under Assumption 1, we can turn Lemma 1 into an explicit upper bound in terms of pairwise CS divergences.

**Proposition 1.** *Let $P_i^c := P(Z \mid Y = c, D = e_i)$ and assume Assumption 1 holds for every class $c$. Then there exist positive constants $C_1$ and $C_2$, depending only on the density-ratio bounds $\{a_c, b_c\}$ and the per-class barycenters, such that*

$$I(Z; D \mid Y) = \mathbb{E}_{c \sim P(Y)}\Big[\mathrm{JS}\big(\{P_i^c\}_{i=1}^{|\mathcal{E}_s|}\big)\Big] \le C_1 \, \mathbb{E}_{c \sim P(Y)}\left[\frac{1}{|\mathcal{E}_s|} \sum_{i=1}^{|\mathcal{E}_s|} D_{\mathrm{CS}}\big(P_i^c \,\|\, \bar{P}^c\big)\right] \qquad (12)$$

*and*

$$I(Z; D \mid Y) \lesssim \mathbb{E}_{c \sim P(Y)}\left[\sum_{i \ne j} D_{\mathrm{CS}}\big(P_i^c, P_j^c\big)\right], \qquad (13)$$

*where the notation $\lesssim$ hides the positive constant $C_2$. The proof proceeds in three steps: (i) Lemma 1 expresses $I(Z; D \mid Y)$ as an average of per-class JS divergences; (ii) Assumption 1 yields an upper bound of JS by barycentric CS divergences; and (iii) convexity and the concavity of the logarithm relate barycentric CS divergences to the average pairwise CS divergences. A full, measure-theoretic derivation is given in Appendix C.*

While equation 11 provides a clear theoretical target, its direct estimation is infeasible. We therefore construct a chain of tractable surrogates that connects our final training objective to this bound.

Combining the conditional high-probability bound equation 11 with Proposition 1 shows that reducing the pairwise CS divergences $\{D_{\mathrm{CS}}(P_i^c, P_j^c)\}$ for each class $c$ directly tightens a target-relevant generalization bound.

Finally, we connect this theoretical pairwise CS divergence to a practical, sample-based objective. Our algorithm's objective is based on the empirical CS-QMI estimator, $\hat{I}_{\mathrm{CS}}$. The critical insight is that this estimator and the empirical estimator for the CS divergence, $\hat{D}_{\mathrm{CS}}$, are both constructed from the same three underlying quantities: two within-domain kernel similarity sums and one cross-domain kernel similarity sum. Due to their mathematical structure, specifically, the opposing roles

these sums play within their respective logarithmic formulas, maximizing the empirical CS-QMI estimator for the independence between two sets of class-conditional samples is directly coupled with minimizing the empirical CS divergence between them. This establishes that our chosen objective correctly targets the theoretical quantity in the generalization bound at the sample level.

This chain of reasoning demonstrates that minimizing our proposed pairwise training objective serves as a way to tighten the generalization bound. By proposing an objective for a given class $c$ and domain pair $(i, j)$ based on the negative empirical CS-QMI, $-\hat{I}_{CS}(Z^{(i,c)}; Z^{(j,c)})$, we arrive at the final result. Combining these steps yields a high-probability generalization gap controlled directly by our algorithm's objective:

$$P\{|R^t(h) - R^s(h)| \geq \epsilon\} \lesssim \frac{M'}{\epsilon} \sqrt{\sum_{c, i<j} \left(-\hat{I}_{CS}(Z^{(i,c)}; Z^{(j,c)})\right) + C_a} + S, \quad (14)$$

Here, $C_a$ is a constant, $S$ is the source deviation term from equation 6, and $M'$ is the positive constant, defined in Appendix C, collecting terms from the loss function regularity and bounded density ratios. The full theoretical derivation is provided in Appendix C. Equation 14 provides the central theoretical justification for our work: minimizing a pairwise objective based on the negative empirical CS-QMI directly tightens a target-relevant, high-probability bound on out-of-distribution generalization. This objective will be formally defined as our alignment loss in the next section.

## 3 METHODOLOGY: CLASS-CONDITIONAL ALIGNMENT VIA KERNEL CAUCHY-SCHWARZ MI

### 3.1 FROM THE TARGET TO A TRAINABLE LOSS

Our goal is to maximize the class-conditional CS-QMI, $I_{CS}(Z^{(i)}; Z^{(j)}|c)$, between feature representations from any two source domains $(i, j)$ given a class $c$. In practice, we operate on mini-batches of data. Let $P_{\Phi|c}^{(i)}$ and $P_{\Phi|c}^{(j)}$ denote the true (but unknown) feature distributions for a given class $c$ in domains $i$ and $j$, respectively. During training, for each class $c$ present in a mini-batch, we draw sets of feature vectors:

$$Z^{(i,c)} = \{z_a^{(i,c)}\}_{a=1}^m \sim P_{\Phi|c}^{(i)}, \quad Z^{(j,c)} = \{z_b^{(j,c)}\}_{b=1}^m \sim P_{\Phi|c}^{(j)} \quad (15)$$

balanced via random subsampling if needed.

The CS-QMI objective relies on estimating information-theoretic quantities directly from these samples. We employ a non-parametric approach based on Kernel Density Estimation (KDE) (Parzen, 1962). Specifically, we define Information Potentials (IP) (Principe, 2010). First, we compute the Gaussian-kernel Gram matrices $K^{(i,c)}, K^{(j,c)} \in \mathbb{R}^{m \times m}$, where $K_{ab}^{(i,c)} = \kappa_\sigma(z_a^{(i,c)}, z_b^{(i,c)})$. The IP estimators are then given by:

$$\widehat{IP}(Z^{(t,c)}) = \frac{1}{m^2} \sum_{a,b=1}^m \kappa_\sigma(z_a^{(t,c)}, z_b^{(t,c)}) = \frac{1}{m^2} \mathbb{1}^\top K^{(t,c)} \mathbb{1}, \quad \text{where } t \in \{i, j\} \quad (16)$$

where $\mathbb{1}$ is the all-ones vector. This is Information Potential (IP) estimator written in matrix notation, obtained via a Parzen-window (kernel density) plug-in, a core quantity in Information Theoretic Learning (Principe, 2010). The IP has an intuitive physical meaning: it measures the average potential energy of an ensemble of "information particles" (the data samples), where each particle exerts a field defined by the kernel. The terms $\widehat{IP}(X)$ and $\widehat{IP}(Y)$ thus represent the self-potentials of their respective feature sets.

From the definition in equation 9, the plug-in CS divergence estimator can be estimated using IPs as follows,

$$\hat{D}_{CS}(Z^{(i,c)}; Z^{(j,c)}) = \log\left(\widehat{IP}(Z^{(i,c)})\right) + \log\left(\widehat{IP}(Z^{(j,c)})\right) - 2\log\left(\frac{\text{tr}(K^{(i,c)} K^{(j,c)})}{m^2}\right). \quad (17)$$

While minimizing this divergence is the direct objective, a powerful and often more stable way to achieve it is by maximizing the statistical dependence between the corresponding feature random

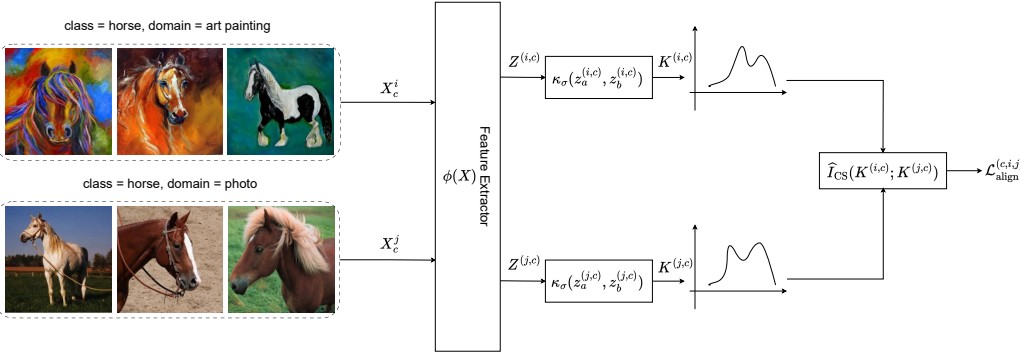

Figure 1: Overview of the proposed DAS-MI framework for class-conditional domain alignment. The process begins with mini-batches of images from two different source domains, domain $i$ (e.g., art painting) and domain $j$ (e.g., photo), that share the same class $c$ (e.g., horse). These image sets, denoted $X_c^i$ and $X_c^j$, are fed into a shared feature extractor $\Phi(X)$ to produce corresponding feature sets $Z^{(i,c)}$ and $Z^{(j,c)}$. For each feature set, a Gaussian-kernel Gram matrix ($K^{(i,c)}$ and $K^{(j,c)}$) is computed, which implicitly represents the feature distribution (visualized as the density plots). These Gram matrices are then used to compute $\hat{I}_{\text{CS}}$, a non-parametric, closed-form estimate of the CS-QMI between the two class-conditional feature distributions. The final alignment loss for this pair, $\mathcal{L}_{\text{align}}^{(i,c,j)}$, is the negative of this estimated mutual information. By minimizing this loss, the model is trained to maximize the statistical dependence between same-class features from different domains, thereby aligning their distributions.

variables. The link is established through mutual information: by constructing pairs of samples (one from each domain) for a given class and enforcing dependence between them, we encourage the feature extractor to map them to similar locations. As this dependence increases, the divergence between their overall (marginal) distributions necessarily decreases. When dependence is maximal (i.e., corresponding samples are mapped to the exact same feature vector), the feature distributions become identical. This motivates reformulating the alignment task as maximizing the CS-QMI, which measures the dependence between the feature random variables of a class $c$ from two different domains $i$ and $j$. Formally, this applies the definition from equation 10 to our feature variables:

$$I_{\text{CS}}(Z^{(i)}; Z^{(j)}|c) = D_{\text{CS}}\big(p(z^{(i,c)}, z^{(j,c)}) \,;\, p(z^{(i,c)})p(z^{(j,c)})\big). \tag{18}$$

Throughout, we form cross-domain, same-class positive pairs: mini-batches from domains $i$ and $j$ are restricted to label $c$, and pairs $(z_a^{(i,c)}, z_b^{(j,c)})$ are drawn within that class. For this protocol, maximizing $I_{\text{CS}}\big(Z^{(i)}; Z^{(j)} \,\big|\, c\big)$ increases cross-domain, same-class dependence and, in turn, decreases equation 17.

### 3.2 ESTIMATING CS-QMI FOR CLASS-DOMAIN PAIRS

A computationally efficient, closed-form estimator for the CS-QMI between the distributions $P_{\Phi|c}^{(i)}$ and $P_{\Phi|c}^{(j)}$, using mini-batches of features $Z^{(i)}$ and $Z^{(j)}$, is given by:

$$\begin{aligned}
\widehat{I}_{\text{CS}}(Z^{(i,c)}; Z^{(j,c)}|c) =& \log\left(\frac{\text{tr}(K^{(i,c)}K^{(j,c)})}{m^2}\right) + \log\left(\widehat{\text{IP}}(Z^{(i,c)})\widehat{\text{IP}}(Z^{(j,c)})\right) \\
& - 2\log\left(\frac{\mathbb{1}^\top K^{(i,c)}K^{(j,c)}\mathbb{1}}{m^3}\right),
\end{aligned} \tag{19}$$

where, in implementation, a small $\epsilon > 0$ is added to the argument of each logarithm for numerical stability. Equation 19 comprises three terms, the joint self-potential $\log\big(\text{tr}(K^{(i,c)}K^{(j,c)})/m^2\big)$, the product of marginal self-potentials $\log\big(\widehat{\text{IP}}(Z^{(i,c)})\,\widehat{\text{IP}}(Z^{(j,c)})\big)$ with $\widehat{\text{IP}}(Z) = m^{-2}\mathbb{1}^\top K\mathbb{1}$, and the cross-potential $-2\log\big(\mathbb{1}^\top K^{(i,c)}K^{(j,c)}\mathbb{1}/m^3\big)$. These three terms correspond to the KDE estimator

---

**Algorithm 1** DAS-MI: Class-Conditional Alignment with Kernel CS-QMI

---

**Require:** Minibatch $\{(x_k, y_k, d_k)\}_{k=1}^{B}$, featurizer $\Phi$, classifier $g$, weight $\lambda_{\text{align}}$, bandwidth $\sigma$.
1: $z_k \leftarrow \Phi(x_k)$, $\hat{y}_k \leftarrow g(z_k)$, $\mathcal{L}_{\text{CE}} \leftarrow \text{CE}(\hat{y}, y)$
2: $\mathcal{L}_{\text{align}} \leftarrow 0$; $\mathcal{C}_{\text{batch}} \leftarrow \text{unique}(\{y_k\})$
3: **for** $c \in \mathcal{C}_{\text{batch}}$ **do**
4:     $\mathcal{D}_c \leftarrow \text{unique}(\{d_k : y_k = c\})$
5:     **for** each $(i, j)$ with $i < j$ in $\mathcal{D}_c$ **do**
6:         $X \leftarrow \{z_k : y_k = c, d_k = i\}$, $Y \leftarrow \{z_k : y_k = c, d_k = j\}$; match to size $m$
7:         Build $K^{(i,c)}, K^{(j,c)}$ with $\kappa_\sigma$; compute $\text{tr}(K^{(i,c)} K^{(j,c)})$, $\mathbb{1}^\top K \mathbb{1}$, $\mathbb{1}^\top K^{(j,c)} \mathbb{1}$, $\mathbb{1}^\top K^{(i,c)} K^{(j,c)} \mathbb{1}$
8:         $\widehat{I}_{\text{CS}} \leftarrow$ calculate using equation 19
9:         $\mathcal{L}_{\text{align}} \leftarrow -\widehat{I}_{\text{CS}}$
10:     **end for**
11: **end for**
12: $\mathcal{L}_{\text{total}} \leftarrow \mathcal{L}_{\text{CE}} + \lambda_{\text{align}} \mathcal{L}_{\text{align}}$;
13: update $(\Phi, g)$

---

of $I_{\text{CS}}$ in equation 10 expressed compactly via traces and quadratic forms (for derivation check Appendix B). In practice we optimize the following alignment loss for each $(c, i, j)$:

$$\mathcal{L}_{\text{align}}^{(c,i,j)} = -\widehat{I}_{\text{CS}}(Z^{(i,c)}; Z^{(j,c)} | c). \tag{20}$$

Summing over present classes and domain pairs yields our training objective:

$$\mathcal{L}_{\text{total}} = \mathcal{L}_{\text{CE}} + \lambda_{\text{align}} \sum_{c \in \mathcal{C}_{\text{batch}}} \sum_{i<j} \mathcal{L}_{\text{align}}^{(c,i,j)}. \tag{21}$$

where $\mathcal{L}_{\text{CE}}$ is the cross entropy loss, and $\lambda_{\text{align}}$ is a scalar hyperparameter that controls the trade-off between the primary classification task (minimizing the cross-entropy loss $\mathcal{L}_{\text{CE}}$) and the alignment objective. The set $\mathcal{C}_{\text{batch}}$ denotes the classes present in the current training batch.

A structurally similar scaffold, supervised ERM with an alignment term, has been widely explored in DG (e.g., moment-based objectives such as MMD and CORAL (Li et al., 2018d; Sun & Saenko, 2016), and gradient-covariance matching, Fish (Shi et al., 2021)). Our objective, however, is materially different: instead of matching moments or gradient statistics, we maximize class-conditional dependence via a closed-form kernel Cauchy-Schwarz QMI (CS-QMI) estimator, explicitly targeting $\widehat{I}_{\text{CS}}(Z^{(i,c)}; Z^{(j,c)} | c)$ to enforce within-class alignment while preserving between-class separation.

The total loss in equation 21 provides a direct, practical mechanism for tightening the theoretical OOD generalization bound established in Section 2.4. The logic is as follows: minimizing our total loss requires minimizing the alignment loss for each class-domain pair (equation 20). By design, this is equivalent to maximizing the empirical CS-QMI between their feature sets. As our theoretical analysis shows, this is the key term controlling the target-domain deviation in our high-probability bound (equation 14). Therefore, each gradient step taken on our final objective directly works to tighten a formal guarantee on OOD performance.

Figure 1 illustrates the end-to-end pipeline of DAS-MI. Mini-batches of images from two different source domains that share the same class are processed by a shared feature extractor, yielding class-conditional feature sets. From these sets, Gaussian-kernel Gram matrices are constructed, implicitly capturing the feature distributions. The CS-QMI between domains is then estimated in closed form from these Gram matrices. Our method reformulates alignment as maximizing statistical dependence between pairs of same-class features across domains. By enforcing such dependence, the feature extractor is encouraged to map cross-domain, same-class samples closer together; as dependence increases, the divergence between their underlying distributions decreases.

The complete training procedure is detailed in Algorithm 1. In practice, for each mini-batch, we iterate through all unique classes and all pairs of domains present for each class. For each such class-domain pair, we gather the feature vectors, compute their kernel Gram matrices, and then use

these matrices to evaluate the closed-form CS-QMI estimator from equation 19. The negative of these CS-QMI values are aggregated to form the total alignment loss, which is weighted by $\lambda_{\text{align}}$ and added to the standard cross-entropy loss to update the model parameters.

# 4 EXPERIMENTAL RESULTS

To empirically validate our proposed method, which we refer to as DAS-MI, we conduct a comprehensive evaluation across five standard domain generalization benchmarks. We analyze its performance under the training-domain validation split model selection protocol and perform ablation studies to understand the key components of our approach.

## 4.1 EXPERIMENTAL SETUP

We evaluate on five DG datasets: **VLCS** (Fang et al., 2013) comprises 5 classes from 4 domains, and 10,729 images in total. **PACS** (Li et al., 2017) contains 9991 images of 7 classes from 4 domains. **OfficeHome** (Venkateswara et al., 2017) contains 15,579 images in total with 65 classes from 4 domains. **TerraIncognita**(Beery et al., 2018) contains 24788 images with 10 classes from 4 domains. **DomainNet** (Peng et al., 2018) is a more recent and the largest dataset which contains 0.6 million images in total with 345 classes from 6 domains. For all benchmarks, we use a ResNet-50 architecture (He et al., 2015) pretrained on ImageNet optimized with Adam optimizer.

In all experiments, the leave-one-domain-out cross-validation protocol is followed. Specifically, in each run a single domain is left out as the target (test) domain, while the rest of the domains are used for training. The final performance of each algorithm is calculated by averaging the top-1 accuracy on the target domain, with different train-validation splits. Our evaluation is built upon the open-source code DomainBed benchmark (Gulrajani & Lopez-Paz, 2021). This ensures fair and reproducible comparisons with state-of-the-art methods. We do not apply early stop strategy for simplicity. For model selection, we select the model checkpoint that performs best on a validation set held out from the training domains. This tests a model's ability to generalize to unseen data while performing the highest on the source domains. The hyperparameters, such as learning rate, weight decay, dropout rate, bandwidth ($\sigma$ used in KDE), and $\lambda_{align}$, are tuned using the sweep script in domainbed and are presented in Table 3. Experiments were run on NVIDIA H100 80GB GPUs, which were portioned into four 20GB instances using Multi-Instance GPU (MIG) technology; the software stack comprised Python 3.11.5 and PyTorch 2.7.1.

## 4.2 COMPARATIVE EVALUATION

We evaluate the DAS-MI model and compare it to state-of-the-art domain-generalization approaches on five standard benchmark datasets, following the common experimental protocol. The comprehensive accuracies (mean ± std) are reported in Table 1. Our DAS-MI method achieves the highest overall average accuracy (66.9 Compared to meta-learning (MLDG (Li et al., 2018a)); data-augmentation strategies (Mixstyle (Zhou et al., 2021), Mixup (Zhang et al., 2017), SagNet (Nam et al., 2019), RSC (Huang et al., 2020), AND-mask (Shahtalebi et al., 2021)); robust-optimization techniques (GroupDRO (Sagawa* et al., 2020), ARM (Zhang et al., 2021), SAM (Foret et al., 2020), GSAM (Zhuang et al., 2022), SAGM (Wang et al., 2023b), Fish (Shi et al., 2021)); and domain-invariant learning methods (MMD (Li et al., 2018b), CORAL (Sun & Saenko, 2016), IRM (Arjovsky et al., 2020), VREx (Krueger et al., 2021)), DAS-MI consistently delivers superior or highly competitive performance. These results suggest that our approach of aligning features on a per-class basis, which preserves core semantic content while simultaneously forcing invariance to domain-specific styles, yields more robust generalization than focusing exclusively on domain-specific or invariant features. Moreover, DAS-MI surpasses the overall average accuracy of the recent GGA (Ballas & Diou, 2025) while surpassing it on two out of five benchmarks.

The results demonstrate the strong capability of DAS_MI to find generalizable solutions when guided by in-distribution validation data. It achieves a strong average accuracy of 66.9% across the five diverse benchmarks. Notably, it achieves state-of-the-art performance on the complex OfficeHome (71.6%) dataset and is highly competitive on the challenging DomainNet (46.9%) and PACS (86.7%) datasets, showcasing its robustness to domain shifts.

Table 1: Comprehensive domain generalization accuracies (mean $\pm$ std). Best in **bold**, second best underlined.

| Algorithm | PACS | VLCS | OfficeHome | TerraInc | DomainNet | Avg |
|---|---|---|---|---|---|---|
| ERM(Vapnik, 2000) | 85.5$\pm$0.2 | 77.5$\pm$0.4 | 66.5$\pm$0.3 | 46.1$\pm$1.8 | 40.9$\pm$0.1 | 63.3 |
| Mixstyle(Zhou et al., 2021) | 85.2$\pm$0.3 | 77.9$\pm$0.5 | 60.4$\pm$0.3 | 44.0$\pm$0.7 | 34.0$\pm$0.1 | 60.3 |
| GroupDRO(Sagawa* et al., 2020) | 84.4$\pm$0.8 | 76.7$\pm$0.6 | 66.0$\pm$0.7 | 43.2$\pm$1.1 | 33.3$\pm$0.2 | 60.7 |
| MMD(Li et al., 2018b) | 84.7$\pm$0.5 | 77.5$\pm$0.9 | 66.3$\pm$0.1 | 42.2$\pm$1.6 | 23.4$\pm$9.5 | 58.8 |
| AND-mask(Shahtalebi et al., 2021) | 84.4$\pm$0.9 | 78.1$\pm$0.9 | 65.6$\pm$0.4 | 44.6$\pm$0.3 | 37.2$\pm$0.6 | 62.0 |
| ARM(Zhang et al., 2021) | 85.1$\pm$0.4 | 77.6$\pm$0.3 | 64.8$\pm$0.3 | 45.5$\pm$0.3 | 35.5$\pm$0.2 | 61.7 |
| IRM(Arjovsky et al., 2020) | 83.5$\pm$0.8 | 78.5$\pm$0.5 | 64.3$\pm$2.2 | 47.6$\pm$0.8 | 33.9$\pm$2.8 | 61.6 |
| MTL(Blanchard et al., 2021) | 84.6$\pm$0.5 | 77.2$\pm$0.4 | 66.4$\pm$0.5 | 45.6$\pm$1.2 | 40.6$\pm$0.1 | 62.9 |
| VREx(Krueger et al., 2021) | 84.9$\pm$0.6 | 78.3$\pm$0.2 | 66.4$\pm$0.6 | 46.4$\pm$0.6 | 33.6$\pm$2.9 | 61.9 |
| MLDG(Li et al., 2018a) | 84.9$\pm$1.0 | 77.2$\pm$0.4 | 66.8$\pm$0.6 | 47.7$\pm$0.2 | 41.2$\pm$0.1 | 63.6 |
| Mixup(Zhang et al., 2017) | 84.6$\pm$0.6 | 77.4$\pm$0.6 | 68.1$\pm$0.3 | 47.9$\pm$0.8 | 39.2$\pm$0.1 | 63.4 |
| SagNet(Nam et al., 2019) | 86.3$\pm$0.2 | 77.8$\pm$0.5 | 68.1$\pm$0.1 | 48.6$\pm$1.0 | 40.3$\pm$0.1 | 64.2 |
| CORAL(Sun & Saenko, 2016) | 86.2$\pm$0.3 | 78.8$\pm$0.6 | 68.7$\pm$0.3 | 47.6$\pm$1.0 | 41.5$\pm$0.1 | 64.5 |
| RSC(Huang et al., 2020) | 85.2$\pm$0.9 | 77.1$\pm$0.5 | 65.5$\pm$0.9 | 46.6$\pm$1.0 | 38.9$\pm$0.5 | 62.7 |
| Fish(Shi et al., 2021) | 85.5$\pm$0.3 | 77.8$\pm$0.3 | 68.6$\pm$0.4 | 45.1$\pm$1.3 | 42.7$\pm$0.2 | 63.9 |
| SAM(Foret et al., 2020) | 85.8$\pm$0.2 | 79.4$\pm$0.1 | 69.6$\pm$0.1 | 43.3$\pm$0.7 | 44.3$\pm$0.0 | 64.5 |
| GSAM(Zhuang et al., 2022) | 85.9$\pm$0.1 | 79.1$\pm$0.2 | 69.3$\pm$0.0 | 47.0$\pm$0.8 | 44.6$\pm$0.2 | 65.1 |
| SAGM(Wang et al., 2023b) | 86.6$\pm$0.2 | **80.0$\pm$0.3** | 70.1$\pm$0.2 | 48.8$\pm$0.9 | 45.0$\pm$0.2 | 66.1 |
| IDM(Dong et al., 2025) | **87.6$\pm$0.3** | 78.1$\pm$0.4 | 68.3$\pm$0.2 | **52.8$\pm$0.5** | 41.8$\pm$0.2 | 65.7 |
| GGA(Ballas & Diou, 2025) | 87.3$\pm$0.4 | 79.9$\pm$0.4 | 68.5$\pm$0.2 | 50.6$\pm$0.1 | 45.2$\pm$0.2 | 66.3 |
| **DAS_MI (Ours)** | 86.7$\pm$0.2 | 79.4$\pm$0.9 | **71.6$\pm$0.1** | 49.7$\pm$0.3 | **46.9$\pm$0.0** | **66.9** |

Table 2: Ablation on the effect of class-conditional alignment. Top-1 accuracy (%) on the five DG benchmarks. "DAS (marginal)" uses the CS-QMI alignment term on marginal feature distributions without conditioning on class labels, while "DAS-MI (class-cond.)" is our full method.

| Method | OfficeHome | VLCS | TerraIncognita | PACS | DomainNet | Avg. |
|---|---|---|---|---|---|---|
| ERM ($\lambda_{align}$=0) | 69.0 | 76.8 | 45.5 | 84.3 | 45.8 | 64.3 |
| DAS (marginal CS-QMI) | 68.8 | 77.0 | 47.5 | 85.6 | 46.4 | 65.1 |
| DAS-MI (class-cond., ours) | 71.6 | 79.4 | 49.7 | 86.7 | 46.9 | 66.9 |

### 4.3 ABLATION: EFFECT OF CLASS-CONDITIONAL CS-QMI ALIGNMENT

We conduct an ablation with three variants: (i) a standard ERM baseline obtained by setting the alignment weight to zero ($\lambda_{align}$=0), which completely removes the CS-QMI term; (ii) a "DAS (marginal)" variant that uses the same CS-QMI estimator and hyperparameters as DAS-MI, but applies alignment to the marginal feature distributions across domains, ignoring class labels; and (iii) the full DAS-MI model with class-conditional CS-QMI, which aligns features only within each class across domains. As shown in Table 2, moving from ERM to marginal CS-QMI yields only a modest average gain, indicating that the estimator alone brings limited benefit when applied at the marginal level. In contrast, enabling class-conditional grouping (DAS-MI) consistently improves performance on all benchmarks and increases the average accuracy to 66.9%.

## 5 CONCLUSION

This paper introduced DAS-MI, a principled approach to domain generalization that addresses the fundamental limitations of marginal feature alignment through class-conditional mutual information maximization. By leveraging the CS-QMI, we derived a closed-form, non-parametric estimator that provides both theoretical guarantees and practical efficiency.

This analytical formulation presents a significant advantage over existing MI-based methods, which typically depend on complex and often unstable techniques like adversarial training or variational approximations. Instead of requiring auxiliary networks or intricate optimization loops, our approach computes the alignment loss directly from kernel Gram matrices. This design not only makes the method computationally efficient and straightforward to integrate into standard training pipelines but also inherently promotes training stability due to the properties of bounded kernel functions.

Empirically, DAS-MI achieved SOTA performance across five standard benchmarks, with particularly strong results on challenging datasets like OfficeHome (71.6%) and DomainNet (46.9%). The consistent improvements across diverse domain shifts validate our hypothesis that preserving class-conditional structure while enforcing cross-domain alignment is crucial for robust generalization.

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

# APPENDIX

- Appendix A: Surveys DG methods including moment matching (e.g., CORAL), MMD-based alignment, risk distribution matching, and MI-based approaches (e.g., MIRO, UFR), and highlights gaps in low-order alignment and lack of class-conditional trainable surrogates.

- Appendix B: Derives the empirical CS-QMI estimator by decomposing information potentials (joint, marginals, cross) under kernel density estimation, yielding a compact Gram-matrix form.

- Appendix C: Provides a class-conditional DG bound linking $I(Z; D \mid Y)$ to pairwise CS divergences, and shows how the trainable CS-QMI objective upper-bounds the conditional mutual information.

- Additional Results (see Fig. 2 and Fig. 3): Presents qualitative t-SNE improvements from class-conditional alignment and kernel bandwidth sensitivity showing robust performance over a wide $\sigma$ range.

- Hyperparameters (see Table 3): Lists training settings per benchmark, including learning rate, regularization, kernel bandwidth, and $\lambda_{\text{align}}$.

## A  RELATED WORK

Domain generalization (DG) has been extensively studied through methods aiming to align distributions across distinct domains. A prominent class of these approaches, known as moment-matching alignment, explicitly matches statistics of latent representations across environments. Methods like Deep CORAL(Sun & Saenko, 2016) achieve alignment by matching the first- and second-order moments of feature distributions across domains, thus reducing distribution shifts. Extending this idea further, Maximum Mean Discrepancy (MMD)-based methods such as MMD-AAE(Li et al., 2018d) and FedKA(Zhang et al., 2024) align all statistical moments within a reproducing-kernel Hilbert space. Despite their effectiveness, these moment-matching methods face significant challenges, particularly the curse of dimensionality, as high-dimensional features are typically sparse and difficult to align accurately. Additionally, aligning raw statistics can lead to matching spurious, domain-specific features that are irrelevant to classification tasks.

To mitigate dimensionality issues associated with representation matching, alternative methods like Risk Distribution Matching (RDM) have been proposed(Nguyen et al., 2024). Instead of aligning latent representations directly, RDM matches scalar distributions of per-sample risks across domains. By minimizing the MMD between the worst-case domain's risk distribution and the aggregated risk from all other domains, RDM achieves robust invariance efficiently. Empirical studies show that RDM consistently outperforms moment-matching methods such as CORAL, and gradient-based alignment approaches.

Another recent avenue leverages mutual information for domain alignment, as information theory provides rigorous guarantees for generalization. Methods such as MIRO(Cha et al., 2022) maximize mutual information between learned features and oracle (pre-trained) representations, guiding encoders toward domain-agnostic features while simultaneously minimizing mutual information with domain labels. Similarly, the Uniformly Distributed Feature Representations (UFR) method(Krishnamachari et al., 2024) enforces a uniform distribution across class-conditioned features, flattening the mutual information landscape to improve generalization on unseen domains. These approaches use MI either as an anchor-to-representation preservation objective or as a uniformity-inducing regularizer; they differ in the MI target and typically rely on auxiliary networks or priors to realize the estimator.

A recent line of work provides a formal, information-theoretic grounding for domain generalization. Notably, (Dong et al., 2025) establish high-probability generalization bounds for unseen target domains, showing that the domain-level generalization gap is controlled by the mutual information $I(Z; D)$ between the learned representation $Z$ and the domain identifier $D$. Building on this analysis, they introduce Inter-Domain Distribution Matching (IDM) with a Per-sample Distribution Matching (PDM) penalty: a practical training recipe that aligns feature and gradient distributions across source domains. IDM/PDM operates at the domain-marginal, label-agnostic level and can over-align nuisance variation when class-conditional structure differs across domains; it also adds overhead from per-dimension matching.

The literature above reveals two gaps: (i) moment/gradient matching focuses on low-order statistics that may over-align nuisance variation, and (ii) information-theoretic bound at the feature level without an explicit, trainable class-conditional surrogate. Our approach addresses these by aligning within-class, cross-domain dependencies using a kernel CS-QMI objective with a tractable matrix

form. Operating directly on pairwise, class-conditional feature sets provides higher-order sensitivity than moment matching and avoids adversarial training or heavy auxiliary networks typical in MI-based estimators. We later show that the resulting objective serves as a practical surrogate that tightens a class-conditional analogue of the domain-level bounds in(Dong et al., 2025).

We do not claim that "feature matching" is new. Our contribution is to instantiate a class-conditional, kernel CS-QMI alignment objective that (i) targets within-class, cross-domain dependence beyond low-order moments, (ii) admits a closed-form, non-adversarial, non-parametric estimator implementable with Gram matrices, and (iii) links to a Rényi divergence between joint and product-of-marginals, clarifying what is controlled by training. This positions our method as a distributional dependence matcher with higher-order sensitivity rather than a moment-only matcher.

## B  DERIVATION OF THE CS-QMI ESTIMATOR

This section provides a detailed derivation for the empirical estimator of CS-QMI, as presented in equation 19 of the main text. The derivation begins with the formal definition of CS-QMI and uses the principles of Information Theoretic Learning, particularly nonparametric density estimation with kernels, to arrive at the final matrix-based formula.

### B.1  DECOMPOSITION OF CS-QMI INTO INFORMATION POTENTIALS

The CS-QMI is defined as the Cauchy-Schwarz divergence between the joint probability distribution $p(z^{(i)}, z^{(j)})$ and the product of its marginals $p(z^{(i)})p(z^{(j)})$. Using the definition from equation 10 and the properties of logarithms, we can expand the expression into three distinct information potential terms:

$$I_{CS}(Z^{(i)}; Z^{(j)}) = D_{CS}(p(z^{(i)}, z^{(j)}); p(z^{(i)})p(z^{(j)}))$$

$$= -\log\left(\frac{\left(\int p(z^{(i)}, z^{(j)})p(z^{(i)})p(z^{(j)})dz^{(i)}dz^{(j)}\right)^2}{\int p(z^{(i)}, z^{(j)})^2 dz^{(i)}dz^{(j)} \int (p(z^{(i)})p(z^{(j)}))^2 dz^{(i)}dz^{(j)}}\right) \quad (22)$$

$$= \log(V_J) + \log(V_M) - 2\log(V_C)$$

where the three potential terms are defined as follows:

- **Joint Self-Potential** ($V_J$): The Information Potential (IP) of the joint distribution.

$$V_J = \int p(z^{(i)}, z^{(j)})^2 dz^{(i)}dz^{(j)} \quad (23)$$

- **Product-of-Marginals Self-Potential** ($V_M$): The IP of the product of marginals. This term can be separated into the product of the individual IPs for each marginal distribution.

$$V_M = \int \left(p(z^{(i)})p(z^{(j)})\right)^2 dz^{(i)}dz^{(j)} = \left(\int p(z^{(i)})^2 dz^{(i)}\right)\left(\int p(z^{(j)})^2 dz^{(j)}\right)$$

$$= V(Z^{(i)}) \cdot V(Z^{(j)})$$

- **Cross-Potential** ($V_C$): The Cross-Information Potential (CIP) between the joint distribution and the product of marginals.

$$V_C = \int p(z^{(i)}, z^{(j)})p(z^{(i)})p(z^{(j)})dz^{(i)}dz^{(j)} \quad (24)$$

The estimator for a general PDF $p(z)$ from samples $\{z_k\}_{k=1}^m$ is:

$$\hat{p}(z) = \frac{1}{m}\sum_{k=1}^m \kappa_\sigma(z - z_k) \quad (25)$$

where $\kappa_\sigma$ is a symmetric kernel function, which we take to be the Gaussian kernel. The derivation relies on the following fundamental property of Gaussian kernels.

**Lemma 2** (Gaussian Kernel Product Rule). *For a normalized Gaussian kernel $\kappa_\sigma(x)$, the integral of the product of two kernels centered at different locations has a closed-form solution:*

$$\int \kappa_\sigma(x - x_1)\kappa_\sigma(x - x_2)dx = \kappa_{\sigma\sqrt{2}}(x_1 - x_2) \tag{26}$$

*This result is derived from the convolution property of Gaussian distributions and is used explicitly in the derivation of the Information Potential estimator in (Principe, 2010, Chapter 2, equation 2.13).*

The estimator for $V(Z^{(i)})$ is obtained by plugging the KDE into the integral:

$$
\begin{aligned}
\widehat{V}(Z^{(i)}) &= \int \left(\frac{1}{m}\sum_{a=1}^{m}\kappa_\sigma(z^{(i)} - z_a^{(i)})\right)^2 dz^{(i)} \\
&= \frac{1}{m^2}\sum_{a=1}^{m}\sum_{b=1}^{m}\int \kappa_\sigma(z^{(i)} - z_a^{(i)})\kappa_\sigma(z^{(i)} - z_b^{(i)})dz^{(i)} \\
&= \frac{1}{m^2}\sum_{a=1}^{m}\sum_{b=1}^{m}\kappa_{\sigma\sqrt{2}}(z_a^{(i)} - z_b^{(i)})
\end{aligned}
\tag{27}
$$

Using the Gram matrix $K$ where $K_{ab} = \kappa_\sigma(z_a^{(i)}, z_b^{(i)})$, and for notational simplicity approximating $\kappa_{\sigma\sqrt{2}} \approx \kappa_\sigma$, this becomes the quadratic form $\frac{1}{m^2}\mathbb{1}^T K\mathbb{1}$. Thus, the estimator for $V_M$ is:

$$\widehat{V}_M = \widehat{V}(Z^{(i)})\widehat{V}(Z^{(j)}) = \frac{(\mathbb{1}^T K^{(i)}\mathbb{1})(\mathbb{1}^T K^{(j)}\mathbb{1})}{m^4} \tag{28}$$

This corresponds to the term $\log\left(\widehat{\mathrm{IP}}(Z^{(i,c)})\widehat{\mathrm{IP}}(Z^{(j,c)})\right)$ in equation 19.

For paired data, the joint PDF is estimated from the sample pairs $\{(z_k^{(i)}, z_k^{(j)})\}_{k=1}^{m}$. The estimator for $V_J$ is:

$$
\begin{aligned}
\widehat{V}_J &= \int\int \left(\frac{1}{m}\sum_{k=1}^{m}\kappa_\sigma(z^{(i)} - z_k^{(i)})\kappa_\sigma(z^{(j)} - z_k^{(j)})\right)^2 dz^{(i)}dz^{(j)} \\
&= \frac{1}{m^2}\sum_{k=1}^{m}\sum_{l=1}^{m}\left(\int \kappa_\sigma(z^{(i)} - z_k^{(i)})\kappa_\sigma(z^{(i)} - z_l^{(i)})dz^{(i)}\right) \\
&\quad \left(\int \kappa_\sigma(z^{(j)} - z_k^{(j)})\kappa_\sigma(z^{(j)} - z_l^{(j)})dz^{(j)}\right) \\
&= \frac{1}{m^2}\sum_{k=1}^{m}\sum_{l=1}^{m}\kappa_{\sigma\sqrt{2}}(z_k^{(i)} - z_l^{(i)})\kappa_{\sigma\sqrt{2}}(z_k^{(j)} - z_l^{(j)})
\end{aligned}
\tag{29}
$$

This double summation is equivalent to $\frac{1}{m^2}\sum_{k,l} K_{kl}^{(i)}K_{lk}^{(j)} = \frac{1}{m^2}\mathrm{tr}(K^{(i)}K^{(j)})$. This corresponds to the term $\log\left(\frac{\mathrm{tr}(K^{(i,c)}K^{(j,c)})}{m^2}\right)$ in equation 19.

Finally, we estimate the cross-term by substituting all three KDEs:

$$
\begin{aligned}
\widehat{V}_C &= \int\int \left(\frac{1}{m}\sum_{k}\kappa(z^{(i)} - z_k^{(i)})\kappa(z^{(j)} - z_k^{(j)})\right)\left(\frac{1}{m}\sum_{a}\kappa(z^{(i)} - z_a^{(i)})\right) \\
&\quad \left(\frac{1}{m}\sum_{b}\kappa(z^{(j)} - z_b^{(j)})\right)dz^{(i)}dz^{(j)} \\
&= \frac{1}{m^3}\sum_{k,a,b}\left(\int \kappa(z^{(i)} - z_k^{(i)})\kappa(z^{(i)} - z_a^{(i)})dz^{(i)}\right)\left(\int \kappa(z^{(j)} - z_k^{(j)})\kappa(z^{(j)} - z_b^{(j)})dz^{(j)}\right) \\
&= \frac{1}{m^3}\sum_{k,a,b}\kappa_{\sigma\sqrt{2}}(z_k^{(i)} - z_a^{(i)})\kappa_{\sigma\sqrt{2}}(z_k^{(j)} - z_b^{(j)})
\end{aligned}
\tag{30}
$$

In matrix form, this triple summation is exactly $\frac{1}{m^3}\mathbb{1}^T K^{(i)} K^{(j)} \mathbb{1}$. This corresponds to the term $-2\log\left(\frac{\mathbb{1}^\top K^{(i,c)} K^{(j,c)} \mathbb{1}}{m^3}\right)$ in equation 19.

By substituting these three derived estimators back into the decomposed form of CS-QMI, we arrive at the final, compact matrix expression presented in the main text, which is computationally efficient and grounded in information-theoretic principles.

## C    FULL DERIVATION OF THE CLASS-CONDITIONAL DOMAIN GENERALIZATION BOUND

This section provides the detailed proof of the class-conditional high-probability bound and of Lemma 1 stated in Section 2.4.

### C.1    NOTATION AND ASSUMPTIONS

We first establish the notation and assumptions used throughout the derivation.

**Notation.**

- Domains: $D \in \mathcal{D}$ denotes a domain identifier. We have $m$ observed source domains $D_1, \dots, D_m$ and an unseen target domain $D'$.
- Data: $X \in \mathcal{X}, Y \in \mathcal{Y} = \{1, \dots, C\}$.
- Representation and classifier: $Z = \Phi(X) \in \mathcal{R}$, and $h = g \circ \Phi$.
- Class-conditional feature laws: For class $c$, write $P_i^c := P(Z \mid Y = c, D = i)$. Under uniform weighting, the per-class barycenter is $\bar{P}^c := \frac{1}{m}\sum_{i=1}^m P_i^c$.
- Risks: $R^d(h) := \mathbb{E}_{(X,Y)\sim P(\cdot|D=d)}[\ell(h(X), Y)]$, $R^s := \frac{1}{m}\sum_{i=1}^m R_{D_i}$, and $R := \mathbb{E}_{D\sim\nu} R_D$ for a domain meta-distribution $\nu$.
- CS divergence: $D_{\mathrm{CS}}(p\|q) = \log\int p^2 + \log\int q^2 - 2\log\int pq$.
- CS-QMI estimator: $\widehat{I}_{\mathrm{CS}}(Z^{(i,c)}; Z^{(j,c)} \mid c)$ as defined in equation 19.

**Assumptions.**

*Remark* 1. It is worth noting that disjoint feature supports could theoretically violate Assumption 1. We address this in two parts: 1. Upper Bound ($b_c$): By definition of the barycenter $\bar{P}^c = \frac{1}{m}\sum P_j^c$, the ratio $P_i^c/\bar{P}^c$ is mathematically bounded above, holding unconditionally. 2. Lower Bound ($a_c$): While raw features may be disjoint, our framework operates on Gaussian KDEs, ensuring full support on $\mathbb{R}^d$ and preventing undefined ratios. While $a_c$ may initially be small (loosening the bound), the DAS-MI objective actively maximizes class-conditional overlap, thereby increasing $a_c$ and tightening the generalization guarantee during training.

*Assumption* 2 (Bounded Loss). The loss function $\ell$ is bounded in $[0, 1]$ and its dependence on $Z$ is Lipschitz with constant $\beta$ after the last linear layer of $g$.

*Assumption* 3 (Domain Independence). The target domain is independent of the training process and source domains. The source domains need not be i.i.d. among themselves.

### C.2    SUPPORTING LEMMAS

Lemma 1 stated for a fixed class $c$ with uniform weights over domains $I(Z; D \mid Y = c) = \frac{1}{m}\sum_{i=1}^m D_{\mathrm{KL}}(P_i^c\|\bar{P}^c) = \mathrm{JS}(\{P_i^c\}_{i=1}^m)$

*Proof.* Since $D$ is uniform on $\{1, \dots, m\}$ given $Y = c$ and $P(Z \mid Y = c, D = i) \sim P_i^c$, we have

$$I(Z; D \mid Y = c) = \sum_{i=1}^m \frac{1}{m} \int P_i^c(z) \log \frac{P_i^c(z)}{\bar{p}^c(z)} \, dz = \frac{1}{m}\sum_{i=1}^m D_{\mathrm{KL}}\big(P_i^c\big\|\bar{P}^c\big). \tag{31}$$

By the definition of uniform-weight Jensen-Shannon divergence, $\text{JS}(\{P_i^c\}) = \frac{1}{m}\sum_i D_{\text{KL}}(P_i^c\|\bar{P}^c)$, which gives the result. $\qquad\square$

*Remark* 2. Lemma 1 is presented with uniform domain weights ($P(D|Y) = 1/m$) to simplify the derivation. However, the result extends naturally to general non-uniform weights $\rho_i = P(D = i|Y)$. In this case, the mutual information equates to the generalized Jensen-Shannon divergence, $\text{JS}_\rho(\{P_i^c\}) = \sum \rho_i \text{KL}(P_i^c\|\bar{P}_\rho^c)$, where $\bar{P}_\rho^c$ is the weighted barycenter. Our subsequent bounds rely on Jensen's inequality, which holds for weighted averages, ensuring the theory remains valid for arbitrary domain distributions.

**Lemma 3** (JS-CS Divergence Bound). *Under Assumption 1, which states that for each class $c$, the density ratios $r_i^c(z) := \frac{P_i^c(z)}{\bar{P}^c(z)}$ are bounded as $a_c \le r_i^c(z) \le b_c$:*

$$\text{JS}(\{P_i^c\}) \le C_c \cdot \frac{1}{m}\sum_{i=1}^m D_{\text{CS}}(P_i^c\|\bar{P}^c) \tag{32}$$

*where $C_c = \frac{1}{2a_c\gamma_c}$ for a constant $\gamma_c > 0$ that depends on the envelope parameters and the barycentric density $\bar{P}^c$.*

*Proof.* The proof proceeds in three main steps. First, we establish an upper bound on the Jensen-Shannon (JS) divergence in terms of a quadratic form involving the density ratios. Second, we establish a lower bound on the Cauchy-Schwarz (CS) divergence in terms of a related quadratic form. Finally, we connect these two bounds to prove the main result.

The Jensen-Shannon divergence for a uniform mixture of $m$ distributions is defined as the average KL divergence to the barycenter:

$$\text{JS}(\{P_i^c\}) = \frac{1}{m}\sum_{i=1}^m D_{\text{KL}}(P_i^c\|\bar{P}^c). \tag{33}$$

Let $f(u) = u\log u - u + 1$, such that $D_{\text{KL}}(P_i^c\|\bar{P}^c) = \int \bar{P}^c(z)f(r_i^c(z))\,dz$. The second derivative of $f(u)$ is $f''(u) = 1/u$. By Taylor's theorem with the mean-value form of the remainder around $u = 1$, for any $u \in [a_c, b_c]$, there exists a $\xi_u$ between 1 and $u$ such that:

$$f(u) = f(1) + f'(1)(u-1) + \frac{1}{2}f''(\xi_u)(u-1)^2 = \frac{1}{2\xi_u}(u-1)^2. \tag{34}$$

Since $u \in [a_c, b_c]$ and $\xi_u$ is between 1 and $u$, we have $\xi_u \ge a_c > 0$. This implies $1/\xi_u \le 1/a_c$. Therefore, we have the inequality:

$$f(u) \le \frac{1}{2a_c}(u-1)^2. \tag{35}$$

Integrating this inequality with respect to the measure $\bar{P}^c$ for each domain $i$ yields:

$$D_{\text{KL}}(P_i^c\|\bar{P}^c) = \int \bar{P}^c(z)f(r_i^c(z))\,dz \le \frac{1}{2a_c}\int \bar{P}^c(z)(r_i^c(z)-1)^2\,dz. \tag{36}$$

Averaging over all $m$ domains, we arrive at the upper bound for the JS divergence:

$$\text{JS}(\{P_i^c\}) \le \frac{1}{2a_c}\cdot\frac{1}{m}\sum_{i=1}^m \int \bar{P}^c(z)(r_i^c(z)-1)^2\,dz. \tag{37}$$

The Cauchy-Schwarz divergence is defined as:

$$D_{\text{CS}}(P_i^c\|\bar{P}^c) = -\log\left(\frac{\left(\int P_i^c(z)\bar{P}^c(z)\,dz\right)^2}{\left(\int (P_i^c(z))^2\,dz\right)\left(\int (\bar{P}^c(z))^2\,dz\right)}\right). \tag{38}$$

To simplify this expression, we introduce an inner product space weighted by $d\mu(z) = (\bar{P}^c(z))^2 \, dz$. In this space, the components of the CS divergence become:

$$\int P_i^c \bar{P}^c \, dz = \int r_i^c (\bar{P}^c)^2 \, dz = \langle r_i^c, 1 \rangle_\mu, \tag{39}$$

$$\int (P_i^c)^2 \, dz = \int (r_i^c)^2 (\bar{P}^c)^2 \, dz = \|r_i^c\|_\mu^2, \tag{40}$$

$$\int (\bar{P}^c)^2 \, dz = \int 1^2 (\bar{P}^c)^2 \, dz = \|1\|_\mu^2. \tag{41}$$

The CS divergence can now be written as the negative logarithm of a squared cosine similarity:

$$D_{\mathrm{CS}}(P_i^c \| \bar{P}^c) = -2 \log \left( \frac{\langle r_i^c, 1 \rangle_\mu}{\|r_i^c\|_\mu \|1\|_\mu} \right) = -2 \log \rho_i^c, \quad \text{where } \rho_i^c := \frac{\langle r_i^c, 1 \rangle_\mu}{\|r_i^c\|_\mu \|1\|_\mu}. \tag{42}$$

Using the standard inequality $-\log x \geq 1 - x$ for $x \in (0, 1]$, we get:

$$D_{\mathrm{CS}}(P_i^c \| \bar{P}^c) \geq 2(1 - \rho_i^c). \tag{43}$$

Next, we relate $1 - \rho_i^c$ to the squared distance $\|r_i^c - 1\|_\mu^2$. Consider the expansion:

$$\begin{aligned}
\|r_i^c - 1\|_\mu^2 &= \langle r_i^c - 1, r_i^c - 1 \rangle_\mu \\
&= \|r_i^c\|_\mu^2 - 2\langle r_i^c, 1 \rangle_\mu + \|1\|_\mu^2 \\
&= (\|r_i^c\|_\mu - \|1\|_\mu)^2 + 2\|r_i^c\|_\mu \|1\|_\mu - 2\langle r_i^c, 1 \rangle_\mu \\
&= (\|r_i^c\|_\mu - \|1\|_\mu)^2 + 2\|r_i^c\|_\mu \|1\|_\mu \left( 1 - \frac{\langle r_i^c, 1 \rangle_\mu}{\|r_i^c\|_\mu \|1\|_\mu} \right) \\
&= (\|r_i^c\|_\mu - \|1\|_\mu)^2 + 2\|r_i^c\|_\mu \|1\|_\mu (1 - \rho_i^c).
\end{aligned} \tag{44}$$

Rearranging this exact identity gives:

$$1 - \rho_i^c = \frac{\|r_i^c - 1\|_\mu^2 - (\|r_i^c\|_\mu - \|1\|_\mu)^2}{2\|r_i^c\|_\mu \|1\|_\mu}. \tag{45}$$

Substituting this back into equation 43:

$$D_{\mathrm{CS}}(P_i^c \| \bar{P}^c) \geq 2 \left( \frac{\|r_i^c - 1\|_\mu^2 - (\|r_i^c\|_\mu - \|1\|_\mu)^2}{2\|r_i^c\|_\mu \|1\|_\mu} \right) = \frac{\|r_i^c - 1\|_\mu^2}{\|r_i^c\|_\mu \|1\|_\mu} - \frac{(\|r_i^c\|_\mu - \|1\|_\mu)^2}{\|r_i^c\|_\mu \|1\|_\mu}. \tag{46}$$

Since the second term is non-negative, dropping it provides a valid lower bound:

$$D_{\mathrm{CS}}(P_i^c \| \bar{P}^c) \geq \frac{\|r_i^c - 1\|_\mu^2}{\|r_i^c\|_\mu \|1\|_\mu}. \tag{47}$$

We now relate the quadratic form in the $\mu$-norm, $\|r_i^c - 1\|_\mu^2 = \int (\bar{P}^c)^2 (r_i^c - 1)^2 \, dz$, to the quadratic form from Step 1, $\int \bar{P}^c (r_i^c - 1)^2 \, dz$. There must exist a constant $\underline{\kappa}_c > 0$ such that $\inf_{z \in \mathrm{supp}(\bar{P}^c)} \bar{P}^c(z) \geq \underline{\kappa}_c$. This implies $(\bar{P}^c(z))^2 \geq \underline{\kappa}_c \bar{P}^c(z)$, which gives:

$$\|r_i^c - 1\|_\mu^2 = \int (\bar{P}^c)^2 (r_i^c - 1)^2 \, dz \geq \underline{\kappa}_c \int \bar{P}^c (r_i^c - 1)^2 \, dz. \tag{48}$$

Furthermore, the denominator term $\|r_i^c\|_\mu \|1\|_\mu$ can be bounded from above. Since $r_i^c \leq b_c$:

$$\|r_i^c\|_\mu^2 = \int (r_i^c)^2 (\bar{P}^c)^2 \, dz \leq b_c^2 \int (\bar{P}^c)^2 \, dz = b_c^2 \|1\|_\mu^2. \tag{49}$$

This means $\|r_i^c\|_\mu \|1\|_\mu \leq b_c \|1\|_\mu^2 = b_c \int (\bar{P}^c)^2 \, dz$. Let us define $M_c = b_c \int (\bar{P}^c)^2 \, dz$, which is a finite constant for class $c$. Substituting these bounds into equation 47:

$$D_{\mathrm{CS}}(P_i^c \| \bar{P}^c) \geq \frac{\underline{\kappa}_c \int \bar{P}^c (r_i^c - 1)^2 \, dz}{M_c} = \gamma_c \int \bar{P}^c(z)(r_i^c(z) - 1)^2 \, dz, \tag{50}$$

where we define the positive constant $\gamma_c := \underline{\kappa}_c/M_c$. Rearranging gives us a bound on the quadratic form:

$$\int \bar{P}^c(z)(r_i^c(z) - 1)^2 \, dz \leq \frac{1}{\gamma_c} D_{\text{CS}}(P_i^c \| \bar{P}^c). \tag{51}$$

Finally, we substitute this result into our JS divergence bound from equation 37:

$$\text{JS}(\{P_i^c\}) \leq \frac{1}{2a_c} \cdot \frac{1}{m} \sum_{i=1}^{m} \left( \frac{1}{\gamma_c} D_{\text{CS}}(P_i^c \| \bar{P}^c) \right). \tag{52}$$

By defining the constant $C_c = \frac{1}{2a_c\gamma_c}$, we arrive at the desired result:

$$\text{JS}(\{P_i^c\}) \leq C_c \cdot \frac{1}{m} \sum_{i=1}^{m} D_{\text{CS}}(P_i^c \| \bar{P}^c). \qquad \square$$

**Lemma 4** (Barycentric to Pairwise CS Bound). *For each class c, let $\bar{P}^c = \frac{1}{m} \sum_{j=1}^{m} P_j^c$ be the barycenter of the class-conditional distributions. Then, the average barycentric CS divergence is bounded by the average pairwise CS divergence:*

$$\frac{1}{m} \sum_{i=1}^{m} D_{\text{CS}}(P_i^c \| \bar{P}^c) \leq \frac{1}{m^2} \sum_{i \neq j}^{m} D_{\text{CS}}(P_i^c \| P_j^c). \tag{53}$$

*Proof.* We provide a detailed proof using the convexity of quadratic forms and the concavity of the logarithm.

The Cauchy-Schwarz divergence between two densities $p$ and $q$ is defined as:

$$D_{\text{CS}}(p\|q) = \log \left( \int p^2 \right) + \log \left( \int q^2 \right) - 2\log \left( \int pq \right). \tag{54}$$

We apply this definition to the left-hand side (LHS) of the lemma's inequality, where $p = P_i^c$ and $q = \bar{P}^c$:

$$\frac{1}{m} \sum_{i=1}^{m} D_{\text{CS}}(P_i^c \| \bar{P}^c) = \frac{1}{m} \sum_{i=1}^{m} \left[ \log \left( \int (P_i^c)^2 \right) + \log \left( \int (\bar{P}^c)^2 \right) - 2\log \left( \int P_i^c \bar{P}^c \right) \right]$$
$$= \left( \frac{1}{m} \sum_{i=1}^{m} \log \left( \int (P_i^c)^2 \right) \right) + \log \left( \int (\bar{P}^c)^2 \right) - \frac{2}{m} \sum_{i=1}^{m} \log \left( \int P_i^c \bar{P}^c \right). \tag{55}$$

We derive bounds for the two terms involving the barycenter $\bar{P}^c = \frac{1}{m} \sum_j P_j^c$.

First, consider the squared integral term $\log(\int (\bar{P}^c)^2)$. The function $g(P) = \int P^2$ is convex. By Jensen's inequality for convex functions:

$$\int (\bar{P}^c)^2 = \int \left( \frac{1}{m} \sum_{j=1}^{m} P_j^c \right)^2 \leq \frac{1}{m} \sum_{j=1}^{m} \int (P_j^c)^2. \tag{56}$$

Since $\log$ is a monotonically increasing function, we can take the logarithm of both sides. Then, applying Jensen's inequality again for the *concave* log function:

$$\log \left( \int (\bar{P}^c)^2 \right) \leq \log \left( \frac{1}{m} \sum_{j=1}^{m} \int (P_j^c)^2 \right) \leq \frac{1}{m} \sum_{j=1}^{m} \log \left( \int (P_j^c)^2 \right). \tag{57}$$

Second, consider the cross-term $\log(\int P_i^c \bar{P}^c)$. By linearity of the integral and concavity of the log function (Jensen's inequality):

$$\log \left( \int P_i^c \bar{P}^c \right) = \log \left( \int P_i^c \left( \frac{1}{m} \sum_{j=1}^m P_j^c \right) \right) = \log \left( \frac{1}{m} \sum_{j=1}^m \int P_i^c P_j^c \right)$$

$$\geq \frac{1}{m} \sum_{j=1}^m \log \left( \int P_i^c P_j^c \right). \tag{58}$$

Multiplying by $-2/m$ and summing over $i$ reverses the inequality:

$$-\frac{2}{m} \sum_{i=1}^m \log \left( \int P_i^c \bar{P}^c \right) \leq -\frac{2}{m^2} \sum_{i,j=1}^m \log \left( \int P_i^c P_j^c \right). \tag{59}$$

Substitute the bounds from equation 57 and equation 59 into our expanded expression from equation 55:

$$\frac{1}{m} \sum_{i=1}^m D_{\text{CS}}(P_i^c \| \bar{P}^c) \leq \left( \frac{1}{m} \sum_{i=1}^m \log \left( \int (P_i^c)^2 \right) \right) + \left( \frac{1}{m} \sum_{j=1}^m \log \left( \int (P_j^c)^2 \right) \right)$$

$$- \frac{2}{m^2} \sum_{i,j=1}^m \log \left( \int P_i^c P_j^c \right). \tag{60}$$

Since the first two terms are identical (with dummy indices $i$ and $j$), we can combine them:

$$\frac{1}{m} \sum_{i=1}^m D_{\text{CS}}(P_i^c \| \bar{P}^c) \leq \frac{2}{m} \sum_{i=1}^m \log \left( \int (P_i^c)^2 \right) - \frac{2}{m^2} \sum_{i,j=1}^m \log \left( \int P_i^c P_j^c \right). \tag{61}$$

Now we expand the right-hand side (RHS) of the lemma's inequality:

$$\frac{1}{m^2} \sum_{i,j=1}^m D_{\text{CS}}(P_i^c \| P_j^c) = \frac{1}{m^2} \sum_{i,j=1}^m \left[ \log \left( \int (P_i^c)^2 \right) + \log \left( \int (P_j^c)^2 \right) - 2 \log \left( \int P_i^c P_j^c \right) \right]$$

$$= \frac{1}{m^2} \sum_{i,j} \log \left( \int (P_i^c)^2 \right) + \frac{1}{m^2} \sum_{i,j} \log \left( \int (P_j^c)^2 \right) - \frac{2}{m^2} \sum_{i,j} \log \left( \int P_i^c P_j^c \right). \tag{62}$$

For the first term, the summand does not depend on $j$, so $\sum_{j=1}^m \log(\dots) = m \log(\dots)$.

$$\frac{1}{m^2} \sum_{i,j} \log \left( \int (P_i^c)^2 \right) = \frac{1}{m^2} \sum_{i=1}^m m \log \left( \int (P_i^c)^2 \right) = \frac{1}{m} \sum_{i=1}^m \log \left( \int (P_i^c)^2 \right). \tag{63}$$

The second term is identical. Therefore, the average pairwise CS divergence is:

$$\frac{1}{m^2} \sum_{i,j} D_{\text{CS}}(P_i^c \| P_j^c) = \frac{2}{m} \sum_{i=1}^m \log \left( \int (P_i^c)^2 \right) - \frac{2}{m^2} \sum_{i,j} \log \left( \int P_i^c P_j^c \right). \tag{64}$$

This expression exactly matches the upper bound we derived in equation 61. This completes the proof of the main inequality. Finally, because $D_{\text{CS}}(P_i^c \| P_i^c) = 0$, the sum over all pairs $(i,j)$ is equal to the sum over off-diagonal pairs $(i \neq j)$. $\square$

**Lemma 5** (CS Divergence to CS-QMI Bound). *For the empirical estimates of class-conditional feature distributions $Z^{(i,c)}$ and $Z^{(j,c)}$:*

$$\widehat{D}_{\text{CS}}(Z^{(i,c)}, Z^{(j,c)}) \leq -\widehat{I}_{\text{CS}}(Z^{(i,c)}; Z^{(j,c)} \mid c) + K_{ij}$$

*where $K_{ij} = 2(\log U_i + \log U_j) + 2 \log m - 3 \log A_{ij}$ with $U_i = \widehat{\text{IP}}(Z^{(i,c)})$ and $A_{ij} = \frac{\text{tr}(K^{(i,c)} K^{(j,c)})}{m^2}$.*

*Proof.* We aim to establish a bound on the pairwise empirical Cauchy-Schwarz (CS) divergence $\widehat{D}_{\text{CS}}$ in terms of the empirical CS-based Quadratic Mutual Information (CS-QMI) $\widehat{I}_{\text{CS}}$. We use the definitions of these estimators (as in Eqs. (11) and (14) of the main paper) and perform algebraic manipulation.

Let's define the following quantities for clarity:

$$U_i := \widehat{\text{IP}}(Z^{(i,c)}) \tag{65}$$

$$A_{ij} := \frac{\text{tr}(K^{(i,c)}K^{(j,c)})}{m^2} \tag{66}$$

$$Q_{ij} := \frac{\mathbf{1}^\top K^{(i,c)}K^{(j,c)}\mathbf{1}}{m^3} \tag{67}$$

Using these notations, the empirical CS divergence and CS-QMI are:

$$\widehat{D}_{\text{CS}}(Z^{(i,c)}, Z^{(j,c)}) = \log U_i + \log U_j - 2\log A_{ij} \tag{68}$$

$$\widehat{I}_{\text{CS}}(Z^{(i,c)}; Z^{(j,c)} \mid c) = \log A_{ij} + \log U_i + \log U_j - 2\log Q_{ij} \tag{69}$$

Our goal is to express $\widehat{D}_{\text{CS}}$ as $\leq -\widehat{I}_{\text{CS}} + K_{ij}$. Let's consider the sum $\widehat{D}_{\text{CS}} + \widehat{I}_{\text{CS}}$:

$$\begin{aligned}
\widehat{D}_{\text{CS}} + \widehat{I}_{\text{CS}} &= (\log U_i + \log U_j - 2\log A_{ij}) + (\log A_{ij} + \log U_i + \log U_j - 2\log Q_{ij}) \\
&= 2(\log U_i + \log U_j) - \log A_{ij} - 2\log Q_{ij}
\end{aligned} \tag{70}$$

Next, we establish an inequality relating $Q_{ij}$ and $A_{ij}$. For Gaussian kernels, Gram matrices $K^{(i,c)}$ and $K^{(j,c)}$ have non-negative entries. It is a known property that if $A$ and $B$ are matrices with non-negative entries, then their product $AB$ also has non-negative entries. Therefore, $X = K^{(i,c)}K^{(j,c)}$ is a matrix with non-negative entries. For any matrix $X$ with non-negative entries, the sum of all its entries is greater than or equal to the sum of its diagonal entries: $\sum_{u,v} X_{uv} \geq \sum_u X_{uu}$.

Applying this to our definitions:

$$\begin{aligned}
\mathbf{1}^\top K^{(i,c)}K^{(j,c)}\mathbf{1} &= \sum_{u,v}(K^{(i,c)}K^{(j,c)})_{uv} \quad \text{(Sum of all entries of the matrix product)} \\
&\geq \sum_u (K^{(i,c)}K^{(j,c)})_{uu} \quad \text{(Since all entries of } K^{(i,c)}K^{(j,c)} \text{ are non-negative)} \\
&= \text{tr}(K^{(i,c)}K^{(j,c)})
\end{aligned} \tag{71}$$

From the definitions of $A_{ij}$ and $Q_{ij}$: $m^3 Q_{ij} = \mathbf{1}^\top K^{(i,c)}K^{(j,c)}\mathbf{1}$ $m^2 A_{ij} = \text{tr}(K^{(i,c)}K^{(j,c)})$ So the inequality becomes: $m^3 Q_{ij} \geq m^2 A_{ij}$ Dividing by $m^3$ (assuming $m > 0$): $Q_{ij} \geq \frac{1}{m}A_{ij}$

Taking the logarithm of both sides (since $Q_{ij}, A_{ij} > 0$) and multiplying by $-2$ (which reverses the inequality): $-2\log Q_{ij} \leq -2\log\left(\frac{A_{ij}}{m}\right) = -2(\log A_{ij} - \log m) = -2\log A_{ij} + 2\log m$.

Now we substitute this back into the sum $\widehat{D}_{\text{CS}} + \widehat{I}_{\text{CS}}$:

$$\begin{aligned}
\widehat{D}_{\text{CS}} + \widehat{I}_{\text{CS}} &\leq 2(\log U_i + \log U_j) - \log A_{ij} + (-2\log A_{ij} + 2\log m) \\
&= 2(\log U_i + \log U_j) + 2\log m - 3\log A_{ij}
\end{aligned} \tag{72}$$

Rearranging to isolate $\widehat{D}_{\text{CS}}$:

$$\widehat{D}_{\text{CS}}(Z^{(i,c)}, Z^{(j,c)}) \leq -\widehat{I}_{\text{CS}}(Z^{(i,c)}; Z^{(j,c)} \mid c) + (2(\log U_i + \log U_j) + 2\log m - 3\log A_{ij}). \tag{73}$$

Let $K_{ij} = 2(\log U_i + \log U_j) + 2\log m - 3\log A_{ij}$. This yields the desired bound:

$$\widehat{D}_{\text{CS}}(Z^{(i,c)}, Z^{(j,c)}) \leq -\widehat{I}_{\text{CS}}(Z^{(i,c)}; Z^{(j,c)} \mid c) + K_{ij}. \qquad \square$$

### C.3 MAIN DERIVATION

**Theorem 1** (Class-Conditional Domain Generalization Bound). *Under Assumptions 2-3, the following bound holds with high probability:*

$$P\{|R^t(h) - R(h)| \geq \varepsilon\} \leq \frac{\sigma'}{\varepsilon}\sqrt{2I(Z; D \mid Y)}$$

*where $I(Z; D \mid Y)$ can be upper bounded by the trainable CS-QMI objectives.*

*Proof of Theorem 1.* We adapt the proof of Theorem 5 from (Dong et al., 2025) by conditioning on the class label $Y$. Starting from the target domain risk gap:

Decompose by conditioning on $Y$:

$$
\begin{aligned}
|R^t(h) - R(h)| &= |\mathbb{E}_Y\left[\mathbb{E}[\ell(h(X), Y)|Y, D = t] - \mathbb{E}[\ell(h, Y)|Y]\right]| \\
&\leq \mathbb{E}_Y\left[|\mathbb{E}[\ell|Y, D = t] - \mathbb{E}[\ell|Y]|\right]
\end{aligned}
\tag{74}
$$

where the inequality follows from Jensen's inequality for the convex function.

Apply sub-Gaussian concentration within each class. For fixed $Y = c$, by the bounded loss assumption and following (Dong et al., 2025, Lemma 4):

$$|\mathbb{E}[\ell|Y = c, D = t] - \mathbb{E}[\ell|Y = c]| \leq \sqrt{2\sigma^2 D_{\mathrm{KL}}(P_{Z|Y=c, D=t}\|P_{Z|Y=c})} \tag{75}$$

Taking expectation over $D \sim \nu$ and applying Jensen's inequality for the concave $\sqrt{\cdot}$:

$$\mathbb{E}_D|R^t(h) - R(h)| \leq \sqrt{2\sigma^2\mathbb{E}_{Y,D}[D_{\mathrm{KL}}(P_{Z|Y,D}\|P_{Z|Y})]} \tag{76}$$

Recognize the conditional mutual information:

$$I(Z; D|Y) = \mathbb{E}_{Y,D}[D_{\mathrm{KL}}(P_{Z|Y,D}\|P_{Z|Y})] \tag{77}$$

Apply Markov's inequality:

$$P\{|R^t(h) - R(h)| \geq \varepsilon\} \leq \frac{\mathbb{E}_D|R^t(h) - R(h)|}{\varepsilon} \leq \frac{\sigma'}{\varepsilon}\sqrt{2I(Z; D|Y)} \tag{78}$$

where $\sigma' = \sigma$ inherits the sub-Gaussian parameter from the bounded loss. $\square$

The proof proceeds through a series of lemmas that establish the connection between the information-theoretic quantities and the CS-QMI estimators.

Starting with the definition of conditional mutual information and applying Lemma 1:

$$
\begin{aligned}
I(Z; D \mid Y) &= \sum_c P(Y = c) \cdot I(Z; D \mid Y = c) \\
&= \sum_c P(Y = c) \cdot \mathrm{JS}(\{P_i^c\}_{i=1}^m)
\end{aligned}
\tag{79}
$$

Next, we apply the JS-CS bound from Lemma 3 to replace the Jensen-Shannon divergence with the average barycentric CS divergence:

$$I(Z; D \mid Y) \leq \sum_c P(Y = c) \cdot \left(C_c \cdot \frac{1}{m}\sum_{i=1}^m D_{\mathrm{CS}}(P_i^c\|\bar{P}^c)\right) \tag{80}$$

Finally, we apply Lemma 4 to replace the average barycentric CS divergence with the average pairwise CS divergence. Note that the original lemma has a factor of $1/2$ for the sum over $i \neq j$, which

is equivalent to summing over all pairs $(i, j)$ and dividing by 2. For simplicity in the chain, we use the sum over all pairs, which is equivalent for our purposes.

$$I(Z; D \mid Y) \leq \sum_c P(Y = c) \cdot C_c \cdot \left( \frac{1}{m^2} \sum_{i,j=1}^m D_{\mathrm{CS}}(P_i^c \| P_j^c) \right)$$

$$= \frac{1}{m^2} \sum_c P(Y = c) C_c \sum_{i,j=1}^m D_{\mathrm{CS}}(P_i^c \| P_j^c) \tag{81}$$

This equation provides a crucial intermediate result: the conditional mutual information is upper-bounded by a weighted average of *population-level* pairwise CS divergences.

Our training objective involves empirical estimators, not the true population divergences. To bridge this gap, we decompose the population divergence $D_{\mathrm{CS}}$ into its empirical counterpart $\widehat{D}_{\mathrm{CS}}$ and an estimation error term $\mathcal{E}_{ij}^c$:

$$D_{\mathrm{CS}}(P_i^c \| P_j^c) = \widehat{D}_{\mathrm{CS}}(Z^{(i,c)}, Z^{(j,c)}) + \mathcal{E}_{ij}^c \tag{82}$$

where $\mathcal{E}_{ij}^c = D_{\mathrm{CS}}(P_i^c \| P_j^c) - \widehat{D}_{\mathrm{CS}}(Z^{(i,c)}, Z^{(j,c)})$ represents the error in estimating the CS divergence from finite samples. Substituting this into our bound from equation 81:

$$I(Z; D \mid Y) \leq \frac{1}{m^2} \sum_c P(Y = c) C_c \sum_{i,j=1}^m \left( \widehat{D}_{\mathrm{CS}}(Z^{(i,c)}, Z^{(j,c)}) + \mathcal{E}_{ij}^c \right) \tag{83}$$

Now that the bound involves the empirical CS divergence, we can apply Lemma 5, which connects $\widehat{D}_{\mathrm{CS}}$ to our trainable objective, $-\widehat{I}_{\mathrm{CS}}$:

$$\widehat{D}_{\mathrm{CS}}(Z^{(i,c)}, Z^{(j,c)}) \leq -\widehat{I}_{\mathrm{CS}}(Z^{(i,c)}; Z^{(j,c)} \mid c) + K_{ij} \tag{84}$$

Substituting this inequality into our main chain equation 83:

$$I(Z; D \mid Y) \leq \frac{1}{m^2} \sum_c P(Y = c) C_c \sum_{i,j=1}^m \left( -\widehat{I}_{\mathrm{CS}}(Z^{(i,c)}; Z^{(j,c)} \mid c) + K_{ij} + \mathcal{E}_{ij}^c \right) \tag{85}$$

Since $D_{\mathrm{CS}}(P_i^c \| P_i^c) = 0$ and $\widehat{I}_{\mathrm{CS}}$ is typically maximized between different domains, we can restrict the sum to $i \neq j$ and adjust the normalization:

$$I(Z; D \mid Y) \leq \frac{1}{m(m-1)} \sum_c P(Y = c) C_c \sum_{i \neq j} \left( -\widehat{I}_{\mathrm{CS}}(Z^{(i,c)}; Z^{(j,c)} \mid c) + K_{ij} + \mathcal{E}_{ij}^c \right) \tag{86}$$

Finally, we substitute this comprehensive upper bound for $I(Z; D \mid Y)$ back into the high-probability bound from Theorem 1:

$$P\{|R^t(h) - R(h)| \geq \varepsilon\} \leq \frac{\sigma'}{\varepsilon} \sqrt{2 I(Z; D \mid Y)} \tag{87}$$

This yields:

$$P\{|R^t(h) - R(h)| \geq \varepsilon\}$$
$$\leq \frac{\sigma'}{\varepsilon} \sqrt{\frac{2}{m(m-1)} \sum_{c, i \neq j} P(Y = c) C_c \left( -\widehat{I}_{\mathrm{CS}}(Z^{(i,c)}; Z^{(j,c)} \mid c) + K_{ij} + \mathcal{E}_{ij}^c \right)} \tag{88}$$

We can group the terms inside the square root to clarify their roles:

$$\cdots \leq \frac{\sigma'}{\varepsilon} \sqrt{\underbrace{\sum_{c, i \neq j} W_{ij}^c \left( -\widehat{I}_{\mathrm{CS}}(Z^{(i,c)}; Z^{(j,c)} \mid c) \right)}_{\text{Trainable Objective}} + \underbrace{\sum_{c, i \neq j} W_{ij}^c K_{ij}}_{\text{Data-dependent Constant}} + \underbrace{\sum_{c, i \neq j} W_{ij}^c \mathcal{E}_{ij}^c}_{\text{Estimation Error}}} \tag{89}$$

where $W_{ij}^c = \frac{2P(Y=c)C_c}{m(m-1)}$ are positive weights.

Let

$$W_{\max} := \max_{c,\,i \neq j} W_{ij}^c \quad \text{and} \quad M' := \sigma' \sqrt{2W_{\max}}. \tag{90}$$

Since $W_{ij}^c = \frac{2P(Y=c)C_c}{m(m-1)}$, we have

$$W_{\max} \leq \frac{2C_{\max}}{m(m-1)}, \qquad C_{\max} := \max_c C_c, \tag{91}$$

so $M'$ depends only on the loss regularity constant $\sigma'$ from Assumption 2 and on the density-ratio envelope parameters encoded in $\{C_c\}_c$.

Using $W_{ij}^c \leq W_{\max}$ and absorbing the weighted sums of $K_{ij}$ and $\mathcal{E}_{ij}^c$ into a finite constant $C_a$, the bound above can be written, up to universal multiplicative constants, in the form

$$P\{|R^t(h) - R(h)| \geq \varepsilon\} \lesssim \frac{M'}{\varepsilon} \sqrt{\sum_{c,\,i \neq j} \left(-\widehat{I}_{\mathrm{CS}}(Z^{(i,c)}; Z^{(j,c)} \mid c)\right) + C_a}. \tag{92}$$

Finally, combining this bound with the triangle inequality in equation 4 to reintroduce the source deviation term $S := P\{|R^s(h) - R(h)| \geq (1-\lambda)\epsilon\}$ from equation 6 yields Eq. equation 14 in the main text:

$$P\{|R^t(h) - R^s(h)| \geq \epsilon\} \lesssim \frac{M'}{\epsilon} \sqrt{\sum_{c,\,i < j} \left(-\widehat{I}_{\mathrm{CS}}(Z^{(i,c)}; Z^{(j,c)})\right) + C_a} + S. \tag{93}$$

This makes explicit that $M'$ collects exactly the dependence on the bounded loss assumption (through $\sigma'$) and on the bounded density ratios (through the constants $C_c$), as stated below Eq. equation 14.

# D  ADDITIONAL RESULTS

## D.1  EVALUATION OF DAS-MI IMPACT

To visually demonstrate the impact of our class-conditional alignment, we compare two models on the PACS benchmark, holding out the "sketch" domain as the unseen target. The first model is a standard Empirical Risk Minimization (ERM) baseline trained only with the cross-entropy loss $\mathcal{L}_{\mathrm{CE}}$. The second is our full DAS-MI model, which incorporates the alignment objective from equation 21 to maximize class-conditional CS-QMI across source domains. Figure 2 presents a t-SNE visualization of the learned feature embeddings $Z$ from both models. The ERM baseline (left) produces diffuse, domain-biased clusters with considerable overlap between classes. In stark contrast, the full DAS-MI model (right) learns features that form tight, well-separated clusters for each class, where samples from different source domains are seamlessly intermixed. This qualitative improvement in cluster cohesion directly illustrates the effectiveness of our alignment term in reducing domain-induced feature discrepancy.

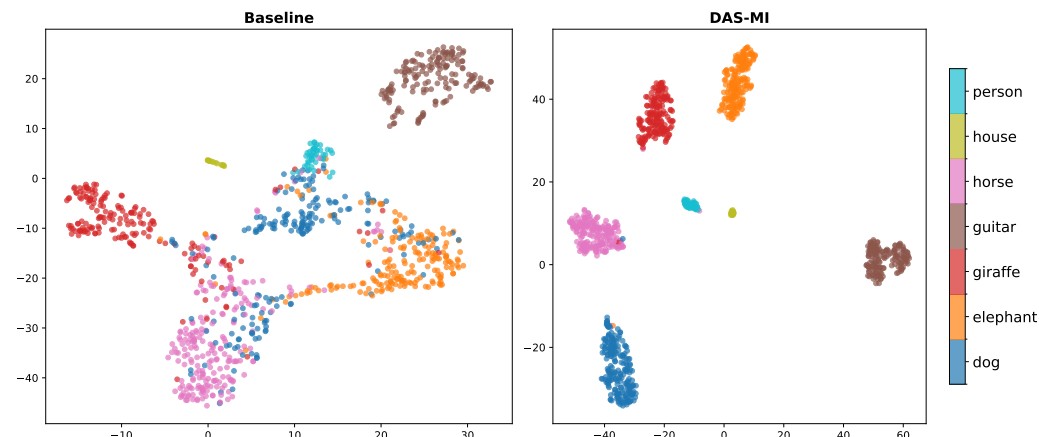

Figure 2: t-SNE (van der Maaten & Hinton, 2008) of the learned feature representation $Z$ on the PACS benchmark (target domain: sketch). The full model (right) includes the alignment term $\mathcal{L}_{\text{align}}^{(c,i,j)}$ from equation 21, whereas the baseline (ERM) in the left omits this term.

### D.2 SENSITIVITY ANALYSIS

A key hyperparameter in our method is the bandwidth $\sigma$ of the Gaussian kernel, which controls the scale at which feature similarity is measured in the alignment loss. An overly sensitive model would require careful, expensive tuning of $\sigma$ for each new task. To assess the robustness of DAS-MI, we conducted a sensitivity analysis, the results of which are shown in Figure 3. We varied $\sigma$ over a wide range while holding all other hyperparameters constant and measured the average test accuracy on five benchmarks.

The results clearly indicate that our method is not overly sensitive to this parameter. For the PACS, VLCS, and OfficeHome datasets, performance remains consistently high and stable across the entire tested range. While more challenging datasets like TerraIncognita and DomainNet show minor fluctuations, the performance does not degrade catastrophically, indicating a broad region of effective $\sigma$ values. This stability is a significant practical advantage, suggesting that a reasonable default value for $\sigma$ can achieve strong performance, thereby simplifying the tuning process and enhancing the reliability of our approach.

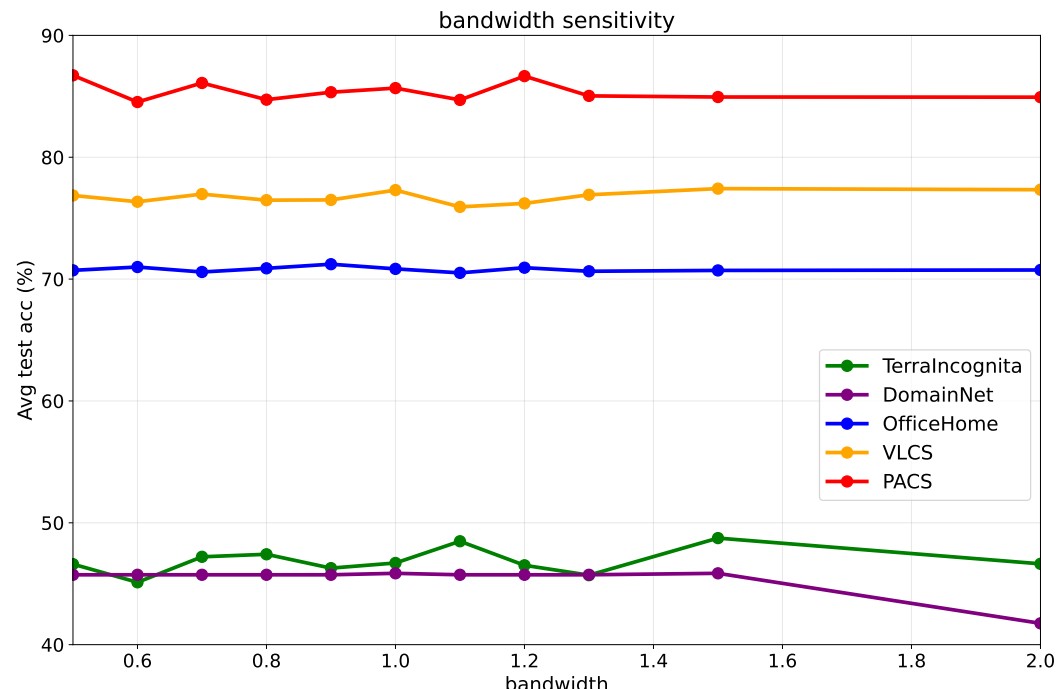

Figure 3: **Sensitivity to Kernel Bandwidth ($\sigma$).** We evaluate the performance of DAS-MI across five benchmarks while varying the Gaussian kernel bandwidth, $\sigma$, and keeping all other hyperparameters fixed. The plots show that for most datasets, particularly PACS, VLCS, and OfficeHome, the model's accuracy is remarkably stable across a wide range of $\sigma$ values (from 0.5 to 2.0). This demonstrates that our method is robust to this key hyperparameter, reducing the need for extensive, dataset-specific tuning.

# E  HYPERPARAMETERS

Table 3 lists the hyperparameters used to produce the results reported in Table 1 across PACS, VLCS, OfficeHome (OH), TerraIncognita (TI), and DomainNet (DN), where learning rate sets the optimizer step size; dropout and weight decay provide regularization; training steps and batch size determine the update schedule; min samples is a batching control to ensure sufficient examples for stable class-conditional alignment; the kernel bandwidth sets the Gaussian scale used by the CS-QMI estimator; and $\lambda_{align}$ balances the cross-entropy loss $\mathcal{L}_{CE}$) with the CS-QMI alignment objective.

Table 3: Hyperparameters for DG experiments. OH, TI, and DN stand for OfficeHome, TerraIncognita, and DomainNet respectively. $\lambda_{align}$ is a scalar hyperparameter that controls the trade-off between the primary classification task (minimizing the cross-entropy loss $\mathcal{L}_{CE}$) and the alignment objective.

| Hyperparameter | PACS | VLCS | OH | TI | DN |
|---|---|---|---|---|---|
| Learning rate | 2.5e-5 | 2.5e-5 | 5e-5 | 2.5e-5 | 2.5e-5 |
| Dropout | 0.5 | 0.5 | 0 | 0.5 | 0.5 |
| Weight decay | 2.25e-5 | 2.25e-5 | 0 | 2.25e-5 | 2.25e-5 |
| Training Steps | 5000 | 5000 | 5000 | 5000 | 5000 |
| Batch size | 64 | 40 | 40 | 40 | 64 |
| min samples | 4 | 4 | 4 | 4 | 4 |
| Bandwidth | 1.0 | 1.5 | 1.5 | 1.5 | 1.0 |
| $\lambda_{align}$ | 0.3 | 0.15 | 0.15 | 0.15 | 0.3 |

## STATEMENT ON THE USE OF LARGE LANGUAGE MODELS (LLMS)

We used Large Language Models (LLMs) as assistive tools in two ways: (i) polishing author-written text for clarity and style, and (ii) suggesting search keywords and potential related work. All technical content, proofs, methods, experiments, and conclusions were conceived, implemented, and written by the authors.

