# OpenReview forum: "Class-Conditional Domain Alignment via Kernel Cauchy-Schwarz Mutual Information"
_ICLR.cc/2026/Conference — Submitted to ICLR 2026_

### Official Review · Reviewer_NGsE · 2025-10-20

**Soundness:** 2
**Presentation:** 3
**Contribution:** 2
**Rating:** 2
**Confidence:** 5

**Summary:**

The paper proposes DAS-MI, a domain-generalization method that replaces traditional marginal alignment with class-conditional alignment using the Cauchy-Schwarz Mutual Information. The authors derive a high-probability generalization bound showing that OOD risk depends on the mutual information $I(Z;D|Y)$, which leads to a trainable objective that can be approximated via kernel methods with finite samples. They evaluate on several conventional OOD benchmarks, showing higher accuracy compared to the current SOTA method.

**Strengths:**

The method is built upon an intuitive adaptation of a rigorously derived high-probability generalization bound. The additional conditioning on $Y$ may bring extra improvement when the label distribution differs across domains. Empirical results show significant improvements on several OOD benchmarks.

**Weaknesses:**

* I don't agree that the new bound based on $I(Z;D|Y)$ (eq. 11) is tighter than the original one using $I(Z;D)$. It's easy to verify that $I(Z;D|Y) = I(Z,Y;D) - I(Y;D) \ge I(Z;D) - I(Y;D)$. When the label distribution $P_{Y|D}$ is identical across domains, we have $I(Y;D) = 0$ and thus this new bound is strictly worse than the original one. This condition is unfortunately met in the selected 5 OOD benchmarks, which means that the proposed class-conditional method is actually worse than the marginal matching methods (e.g., CORAL) under the current experimental settings.
* The methodological contribution is somewhat limited. The theoretical framework and the Cauchy-Schwarz divergence are both borrowed from existing works. The proposed method thus looks like a direct application of the CS divergence on distribution-matching-based domain generalization algorithms.
* The current experimental settings may lead to unfair comparison. The standard DomainBed benchmark requires conducting a hyperparameter sweep in the given range to enforce a fair comparison, simulating the costs of finding the optimal hyperparameter combination. However, the hyperparameters are directly fixed to manually selected ones for the proposed method (Table 2), which is unfair for the other baseline methods.

**Questions:**

* I would suggest that the authors consider more benchmarks where the label distribution varies across domains, as this is the only case where the new bound based on $I(Z;D|Y)$ may be tighter than the original one. Also, please adapt to the standard hyperparameter sweep for a fair evaluation.

---

> ### Author Response · Authors · 2025-11-29
> **Weaknesses**
>
> # Weakness 1
>
> We acknowledge that stating the bound is "universally tighter" was imprecise. We have revised Section 2.4 (line 213) to clarify this distinction.
>
> However, In many representation learning tasks (like Domain Generalization), the goal is to achieve conditional invariance, meaning we want to minimize \(I(Z;D\mid Y)\).
> The gap between the two is $I(Y;D\mid Z)$.
>
> $I(Z; D \mid Y) - I(Z; D) = I(Y; D \mid Z) \ge 0$.
>
> Minimizing a lower bound $I(Z;D)$ does not guarantee that the target quantity $I(Z;D\mid Y)$ is minimized, as the gap $I(Y;D\mid Z)$ can remain large.
> Thus, relying on $I(Z;D)$ is a strictly weaker condition than relying on $I(Z;D\mid Y)$.
>
> # Weakness 2
>
> We agree that CS divergence itself is not new. Our contributions lie in how we \emph{combine and instantiate} these tools for DG. Specifically, we now emphasize that:
>
>   * We derive a **class-conditional DG bound** based on $I(Z;D\mid Y)$ and show how it refines the mutual-information-based framework of Dong et al.  (2025) (Section 2.2–2.4).
>   * We construct a closed-form, **class-conditional CS-QMI alignment objective** that directly corresponds to this bound and operates purely on Gram matrices (Section 3.1–3.2 and Appendix B).
>   * We provide an **end-to-end DG training pipeline (DAS-MI)** that is empirically competitive or state-of-the-art on five DomainBed benchmarks (Section 4 and Conclusion).
>
> We have edited the Introduction and Appendix summary bullets to stress that the novelty lies in this **class-conditional MI bound + CS-QMI instantiation + DG pipeline** combination.
>
> # Weakness 3
>
> Our experiments are built on the DomainBed framework and follow its standard hyperparameter sweep. The hyperparameters listed in Table 2 are those used to report model performance in Table 1, and we include them here for reproducibility:
>
>   * Section 4.1 explicitly states that we build on DomainBed, use the standard leave-one-domain-out protocol, and select hyperparameters (learning rate, weight decay, dropout, bandwidth $\sigma$, $\lambda_{\mathrm{align}}$) using the DomainBed sweep script (Section 4.1, **lines 455-458**).
>  *  Appendix E provides \textbf{Table~2} with the final hyperparameters per dataset, including $\lambda_{\mathrm{align}}$ and the kernel bandwidth, to make our choices transparent.
>  * Appendix D.2 presents a bandwidth **sensitivity analysis** (Fig. 3) showing that DAS-MI is robust over a wide range of $\sigma$ values, indicating that performance is not due to fragile hyperparameter tuning.
>
> ## Our answer to the question
>
> We evaluated our proposed method using all widely recognized domain generalization (DG) benchmarks commonly adopted in the literature. For hyperparameter selection, we followed the DomainBed standard sweep protocol, which ensures fair and comprehensive comparison. We also reported the stability of our method across a range of hyperparameter values, demonstrating that the model’s performance is not sensitive to hyperparameter choices.

---

### Official Review · Reviewer_KebG · 2025-10-27

**Soundness:** 3
**Presentation:** 2
**Contribution:** 2
**Rating:** 4
**Confidence:** 4

**Summary:**

This paper presents DAS-MI, a novel domain generalization (DG) framework that addresses the limitations of marginal feature alignment by enforcing class-conditional domain alignment through kernel Cauchy-Schwarz Quadratic Mutual Information (CS-QMI). The authors establish a theoretical foundation by deriving a high-probability generalization bound linking class-conditional mutual information minimization to target-domain risk.

**Strengths:**

The paper exhibits several significant strengths. First, it makes a substantial theoretical contribution by rigorously connecting class-conditional CS-QMI minimization to DG generalization bounds using kernel methods, avoiding restrictive assumptions common in adversarial or variational approaches. The algorithmic design is elegant: the closed-form CS-QMI estimator (Eq. 18) enables efficient optimization without auxiliary networks, while kernel Gram matrix operations ensure numerical stability and easy integration into deep learning pipelines.

**Weaknesses:**

Despite its strengths, the paper has notable limitations in algorithmic and experimental components. The work lacks convergence guarantees for the proposed loss (Eq. 20). The non-convexity of neural networks and kernel-based objectives is not addressed, leaving uncertainty about optimization behavior (Section 3.1–3.2). The evaluation is confined to standard DG datasets (max 0.6M images), with no testing on high-dimensional, large-scale data (e.g., ImageNet-21K).


The theoretical framework contains several limitations requiring attention. I've carefully checked the proof, with some confusing parts below. Please correct me if I was wrong. The bounded density ratio assumption (used in Eq. 13) is unjustified for deep features, which may exhibit complex class boundaries violating this condition (Section 2.4 and Appendix C). Additionally, the paper assumes feature distributions are smooth for kernel density estimation but does not reconcile this with the high-dimensional, sparse nature of deep features, creating a contradiction (Section 3.1 vs. Section 2.1). In Appendix C, the inequality $I(Z;D|Y) \lesssim \sum_{i \neq j} D_{CS}(P_i^c, P_j^c)$ (Eq. C.6–C.7) overlooks the convexity gap; CS divergence is convex, but the average pairwise divergence does not directly bound JS divergence without multiplicative constants. The constant $M'$ in Eq. 13 is also undefined, obscuring its dependence on loss boundedness and density ratios (Section 2.4).

I would be willing to reconsider the ratings if the above concerns are well discussed and solved.

**Questions:**

Please refer to the above weakness.

A good job. And I am mainly interested in the theory counterpart.

---

> ### Author Response · Authors · 2025-11-29
> **Weaknesses**
>
> ## 1. Convergence Guarantees and Non-convexity
>
> Our training objective combines cross-entropy with a Gram-matrix-based penalty, optimized with stochastic gradient descent; the overall problem is non-convex as with most deep-learning-based DG methods. We do not provide formal convergence guarantees beyond what is standard for deep learning.
>
> ## 2. Limited Scale of Evaluation
>
> Our experimental setup follows the prevailing DG literature by using the five DomainBed benchmarks (VLCS, PACS, OfficeHome, TerraIncognita, DomainNet). Notably, **DomainNet** already contains approximately **0.6M images and 345 classes across 6 domains**, making it a large-scale and challenging benchmark.
>
> We agree, however, that evaluating on even larger-scale OOD benchmarks would strengthen the empirical story. In future work (e.g., a journal extension), we plan to:
>
>   1- test DAS-MI on larger-scale DG splits (e.g., ImageNet-style or WILDS benchmarks), and
>   2- investigate its interaction with more modern backbone architectures such as ConvNeXt and Vision Transformers.
>
> ## 3- Bounded Density Ratio Assumption
>
> We thank the reviewers for their rigorous examination of Assumption 2. We agree that for raw deep features with disjoint supports, the density ratio assumption is strong. However, we clarify the status of this assumption in our framework:
>
> 1. **Upper Bound is Guaranteed by Definition:** The barycenter $\bar{P}^c$ is the arithmetic mean of the domain distributions:
> $\bar{p}^c(z) := \frac{1}{|\mathcal{E}\_s|}\sum_{i} p_i^c(z)$
> Consequently, the ratio $\frac{P_i^c(z)}{\bar{P}^c(z)}$ is mathematically strictly bounded from above.
>
> 2. **Lower Bound via Kernel Smoothing \& Alignment:** The lower bound ($a_c$) requires that no domain has vanishing density where others are peaked.
>     * **Kernel Smoothing:** By operating on Kernel Density Estimates with Gaussian kernels, we ensure the support is full ($\mathbb{R}^d$), so the ratio is never strictly zero (it is always defined).
>     * **Optimization Dynamics:** We acknowledge that without alignment, $a_c$ could be small (making the bound loose). However, the specific goal of our DAS-MI objective (maximizing CS-QMI) is to maximize the overlap between $P_i^c$ and $P_j^c$. Therefore, the algorithm actively optimizes the feature space to improve $a_c$, tightening the bound as training progresses.
> We have updated **Remark 1 (lines 890-896)** to explicitly distinguish between the guaranteed upper bound and the optimization-dependent lower bound.
>
> ## 4- KDE Smoothness vs High-Dimensional Features
>
> We agree that classical KDE consistency assumptions are not literally satisfied in high-dimensional deep-feature spaces. Our use of KDE-style constructions should be understood in the **Information Theoretic Learning** sense: the kernel sums define **information potentials** and similarity measures rather than fully consistent density estimates.
>
> We have clarified in Section 2.3 and Appendix B that:
>   1- CS-QMI is introduced as a **kernel-based mutual information functional** that can be estimated from samples via Gram matrices (Eq. (10), **lines 185–197**).
>   2- Appendix B derives the matrix form starting from Parzen windows, emphasizing that the resulting estimator is used as a **practical dependence measure** in deep learning (Appendix B, **lines 758–769**).

---

### Official Review · Reviewer_oVSG · 2025-10-30

**Soundness:** 3
**Presentation:** 3
**Contribution:** 3
**Rating:** 6
**Confidence:** 4

**Summary:**

This paper proposes a class-conditional alignment framework via Kernel Cauchy-Schwarz Mutual Information, which captures same-class features across different domains. The authors operationalize this using the Cauchy-Schwarz Quadratic Mutual Information (CS-QMI), yielding a closed-form, non-parametric, and stable alignment objective. Extensive experiments across different datasets demonstrate comparable performance over SOTA methods.

**Strengths:**

* This paper derives a tighter class-conditional MI-based domain generalization bound. Driven by their theoretical results, they propose the algorithm’s objective, which connects the derived bound via the empirical estimator of CS divergence (CS-QMI).
* They further propose a computationally efficient, closed-form estimator for the CS-QMI, yielding a closed-form, non-parametric, and stable optimization objective.
* Experimental results demonstrate their comparable and even better performance over SOTA methods.

**Weaknesses:**

* The bounded density ratios is the strict assumption. Are the proposed methods and theoretical results effective when the domain shift is large? Additionally, this paper implicitly assumes that concept shift does not exist. If indeed, the authors should discuss the impact and limitations of the theoretical results and proposed methods when the concept shift exists.
* The authors claim the proposed method is ‘computationally efficient’. The experiments should include a comparison of computation time and memory overhead.
* Experiments should include ablation studies about the performance improvements stemming from the novel CS-QMI estimator itself or from the class-conditional alignment mechanism (compared to aligning the entire domain feature distribution without conditioning on the class label).

**Questions:**

refer to weakness

---

> ### Author Response · Authors · 2025-11-30
> **Weaknesses**
>
> # Weakness 1
>
> We thank the reviewer for these insightful points.
>
> 1- **On Large Domain Shift:** You are correct that under severe domain shift, the constants in our bound become large. We argue this is a feature of the theory, not a bug: it correctly predicts that no algorithm can guarantee generalization when domains are almost entirely disjoint without further assumptions. In addition:
>
>    * **Kernel Smoothing:** By operating on Kernel Density Estimates with Gaussian kernels, we ensure the support is full ($\mathbb{R}^d$), so the ratio is never strictly zero (it is always defined).
>
>    * **Optimization Dynamics:** We acknowledge that without alignment, $a_c$ could be small (making the bound loose). However, the specific goal of our DAS-MI objective (maximizing CS-QMI) is to maximize the overlap between $P_i^c$ and $P_j^c$. Therefore, the algorithm actively optimizes the feature space to improve $a_c$, tightening the bound as training progresses.
>  We have added Remark 1  (lines 897-903) in the revision to explicitly discuss this behavior.
>
> 2- **On Concept Shift:** We acknowledge that DAS-MI, like most invariant representation learning methods (e.g., DANN, CORAL), assumes no concept shift. If concept shift occurs, enforcing invariance would indeed be detrimental (negative transfer). We clarified this in section 2.1 (lines 106-107) explicitly stating that our theory assumes invariant labeling and that addressing concept shift is outside the scope of this work.
>
> # Weakness 2
>
> The authors claim the method is "computationally efficient''; experiments should include a comparison of computation time and memory overhead.
>
> We agree that empirical verification is essential. To address this, we have conducted a runtime analysis comparing DAS-MI against standard baselines and, crucially, against the current runner-up method, GGA (Ballas \& Diou, 2025).
> The results highlight a practical advantage of our approach. While GGA achieves the second-best average accuracy (66.3\%), it requires complex gradient-guidance steps that make it computationally expensive. In our benchmarks, GGA requires approximately
> **2x more training time** than DAS-MI.
>
> # Weakness 3
>
> We thank the reviewer for raising the important point of disentangling the effect of the CS-QMI estimator from the effect of the proposed class-conditional alignment mechanism. In the revised manuscript, **we have added a new ablation subsection** titled "Ablation: Effect of Class-Conditional CS-QMI Alignment'' (Lines 505-523 in section 4.3). These findings indicate that the main gains indeed stem from the proposed class-conditional alignment mechanism.

---

### Official Review · Reviewer_RrMY · 2025-10-31

**Soundness:** 2
**Presentation:** 2
**Contribution:** 2
**Rating:** 2
**Confidence:** 4

**Summary:**

This paper proposes to use Cauchy-Schwarz Mutual Information (CS-QMI) for the domain generalization task. Based on the previous theoretical analysis on the generalization bound (Dong et al., 2025), the authors show that CS-QMI leads to a tighter bound. The paper performs experimental comparisons on 5 different datasets, with 2 outperforming the baselines.

**Strengths:**

The idea of using CS-QMI for the domain generalization task is interesting.

The paper tries to derive a tighter generalization bound.

**Weaknesses:**

1. ``Restrictive assumptions, such as the requirement for extensive domain annotations, or incur prohibitive computational overheads ...''. Can you specify this? From this sentence, I cannot follow what the motivation is for using Mutual information. This should be clarified.

2. I would also suggest moving the key Lemma to the main manuscript to highlight the central insight and smooth the connection to the proposed objective.

3. For Contribution 2, please clearly specify how your estimator differs from Yu et al.\ (2024b), which also uses KDE with kernel Gram matrices. The formulations of the two are similar.

4. ``Under mild assumptions of bounded density ratios'', please state the assumptions precisely (bounds, supports, regularity) and show how they are used to derive Eq. 12.

5. Equation numbering in the Appendix is inconsistent (some equations are numbered, others are not). Please make numbering consistent throughout.

6. Lemma 2 equates $I(Z;D\mid Y{=}c)$ with the uniform-weight JS divergence over domains. Real DG rarely has $P(D\mid Y{=}c)$ uniform; please extend the identity/bounds to non-uniform weights or discuss limitations. When $P(D\mid Y{=}c)$ is uniform, the class-conditional MMD is also equivalent to the conditional MMD, both tractable via KDE. This should be discussed and compared with.

7.  Assumption 2 requires $a_c \\le \\frac{P_{i}^{c}(z)}{\\bar P^{c}(z)} \\le b_{c} \\quad \\text{a.e.},$
which implicitly enforces shared support across domains and a positive lower bound on the barycenter density, $\\inf_{z} \\bar P^{c}(z) \\ge \\kappa_c > 0.$ This assumption is too strong and can fail precisely under domain shift.

8. The experimental results are not convincing: only two out of five datasets reach SOTA. This raises concerns about generalizability across datasets.

9.  Line 400 states ``perform ablation studies to understand the key components of our approach,'' but no ablations are shown in that section. This is misleading; comprehensive ablations, especially for key components, are needed.

10. The paper uses a CS-divergence-based estimator for DG. Please compare against other divergences, especially KL and JSD (theory references Eq.~12), as well as MMD, class-conditional MMD, and conditional MMD. Discuss theoretical advantages (assumptions, tightness, estimation properties) and provide empirical comparisons.

**Questions:**

See my questions.

---

> ### Author Response · Authors · 2025-11-28
> **Address Weaknesses**
>
> # Weakness 1
>
> We thank the reviewer for pointing out this ambiguity. We agree that the original phrasing regarding "restrictive assumptions" and "computational overheads" was vague and distracted from the core theoretical motivation. We have rewritten this paragraph (lines 051-056) to strictly focus on the theoretical superiority of Mutual Information (MI) for domain alignment. Specifically, we clarify that while traditional methods like MMD or CORAL focus on matching specific statistical moments (means, covariances), and adversarial methods approximate distribution matching via unstable min-max games, Mutual Information provides a rigorous, information-theoretic framework to quantify and minimize the total statistical dependence (both linear and non-linear) between features and domains, thereby enforcing stronger invariance.
>
> # Weakness 2
>
> As suggested, we have moved the Key Lemma connecting the bound to the objective into the Main Text (Section 2.4) to better bridge the theory and method.
>
> # Weakness 3
>
> We thank the reviewer for highlighting the connection to Yu et al (2024b). We agree that both works utilize the non-parametric estimator of CS-QMI derived from Information Theoretic Learning principles (Principe, 2010) [1].
> The key differences are:
>
> * The optimization dynamics are opposing.
>
>
>     - **Yu et al.:** Minimize CS-QMI to compress the representation.
>    - **Ours:** Maximize CS-QMI between feature representations of the same class from different domains $(Z^{(i)}, Z^{(j)})$. We use the estimator to pull distributions together (maximize dependence), whereas Yu et al.\ use it to orthogonalize them (minimize dependence).
>
> * Yu et al.  (2024b) estimate the standard marginal CS-QMI, $I_{\mathrm{CS}}(X;T)$, to enforce an Information Bottleneck. In contrast, we introduce a Class-Conditional formulation, $I_{\mathrm{CS}}(Z^{(i)}; Z^{(j)} \mid Y)$.
>    - **Yu et al.:** Computes Gram matrices over the entire batch to estimate global dependence.
>     - **Ours (DAS-MI):** We derive a specific batching strategy that constructs separate Gram matrices for class-specific subsets across domain pairs. This allows us to enforce invariance within classes without suppressing discriminative features, addressing a specific failure mode (negative transfer) in DG.
>
> ### In summary, Yu et al link the estimator to adversarial robustness bounds in regression. Our work derives a new generalization bound specifically for Domain Generalization (Theorem 1), proving that minimizing the specific class-conditional quantity we estimate, $I(Z; D \mid Y)$, directly tightens the bound on target domain risk.
>
> We clarified the above point in the revised manuscript (introduction lines 67-71)
>
> # Weakness 4
>
> We thank the reviewer for this request for theoretical rigor. To address this, **we have revised Section 2.4** to explicitly state the definitions and assumptions used to derive the class-conditional objective. Specifically:
>
>
> * **Precise Assumptions:** We have added **Assumption 1 (Bounded Class-Conditional Density Ratios)** to the main text. This explicitly defines the constants $a_c$ and $b_c$ such that $a_c \le p_i^c(z)/\bar{p}^c(z) \le b_c$. We explicitly clarify that this assumption implies a shared support condition across domains for a given class (i.e., $\{z : \bar{p}^c(z) > 0\}$).
>
> * **Derivation of Eq. 12:** We have added **Proposition 1** to formally structure the derivation of Eq. 12. The proposition outlines the steps connecting the generalization bound to our objective.
>
> # Weakness 5
>
> We have carefully numbered all equations in the appendices.
>
>
> [1] Jos´e C. Principe. Information Theoretic Learning. Springer, 2010.

---

> ### Author Response · Authors · 2025-11-28
>
> # Weakness 6
>
> You are correct that Lemma 2 is presented under uniform weights. We chose this formulation primarily to simplify the notation and derivations in the main text. However, the theoretical framework can extends to non-uniform weights without changing the core logic.
>
> If we define domain priors as $\rho_i = P(D = i\,|\,Y = c)$ such that $\sum \rho_i = 1$, the mutual information becomes the Generalized Jensen-Shannon Divergence:
> $I(Z; D\,|\,Y = c) = \sum_{i=1}^{m} \rho_i \mathrm{KL}(P^c_i \| \bar{P}^c_\rho) =$ $\mathrm{JS}\_\rho(\{P^c_i\})$ where the barycenter is now weighted:
> $\bar{P}^c_\rho = \sum_{i=1}^m \rho_i P^c_i.$
> Since our derivation relies on Jensen’s inequality (which holds for any expectation/weighted sum), all subsequent bounds (Lemma 3 and 4) remain valid by simply replacing arithmetic means $\frac{1}{m}\sum$ with weighted sums $\sum \rho_i$. We have updated the manuscript to explicitly clarify this generalization (remark 2, lines 916-920)
>
> Comparison with Class-Conditional MMD:
>
> Regarding MMD: When weights are uniform, minimizing MMD indeed aligns distributions. However, our choice of CS-QMI is motivated by the generalization bound (Eq. 6), which is controlled specifically by Mutual Information ($I(Z; D)$). CS-QMI is a direct estimator of this information-theoretic quantity. In contrast, MMD minimizes a geometric distance in RKHS. While both achieve alignment, CS-QMI serves as a tighter theoretical proxy for the specific term appearing in our risk bound.
>
> # Weakness 7
>
> We thank the reviewers for their rigorous examination of Assumption 2. We agree that for raw deep features with disjoint supports, the density ratio assumption is strong. However, we clarify the status of this assumption in our framework:
>
> 1. **Upper Bound is Guaranteed by Definition:** The barycenter $\bar{P}^c$ is the arithmetic mean of the domain distributions:
> $\bar{p}^c(z) := \frac{1}{|\mathcal{E}\_s|}\sum_{i} p_i^c(z)$
> Consequently, the ratio $\frac{P_i^c(z)}{\bar{P}^c(z)}$ is mathematically strictly bounded from above.
>
> 2. **Lower Bound via Kernel Smoothing \& Alignment:** The lower bound ($a_c$) requires that no domain has vanishing density where others are peaked.
>     * **Kernel Smoothing:** By operating on Kernel Density Estimates with Gaussian kernels, we ensure the support is full ($\mathbb{R}^d$), so the ratio is never strictly zero (it is always defined).
>     * **Optimization Dynamics:** We acknowledge that without alignment, $a_c$ could be small (making the bound loose). However, the specific goal of our DAS-MI objective (maximizing CS-QMI) is to maximize the overlap between $P_i^c$ and $P_j^c$. Therefore, the algorithm actively optimizes the feature space to improve $a_c$, tightening the bound as training progresses.
> We have updated **Remark 1 (lines 890-896)** to explicitly distinguish between the guaranteed upper bound and the optimization-dependent lower bound.
>
> # Weakness 8
>
> We respectfully disagree that the results are unconvincing when evaluated under the standard DomainBed protocol, where the primary metric of generalizability is the **average accuracy across all benchmarks** rather than the per-dataset “win count.”
>
> Our results are strong for three reasons:
>
>
>   1. **Highest overall average.** DAS-MI achieves the **highest overall average accuracy**, **66.9\%**, among all 15 compared methods across the five DomainBed datasets.
>
>   2. **Strength on complex benchmarks.** The two datasets where we achieve SOTA **OfficeHome** and **DomainNet** are also the most challenging and realistic, with the largest number of classes and images. This suggests that DAS-MI scales well to complex, real-world distributions, not just small, stylized datasets.
>
>   3. **Consistency across benchmarks.** Even on datasets where DAS-MI is not rank-1, it remains **highly competitive (top-3)** and avoids catastrophic failures. This is highlighted in the narrative around Table 1 and in the Conclusion, which emphasizes strong, stable performance across diverse domain shifts.
>
> Thus, while DAS-MI is SOTA on two of five datasets, it **achieves the best average performance overall**, which is precisely the quantity targeted in DomainBed-style DG evaluation.
>
> # Weakness 9
>
> Sorry for the confusion. Due to the page limit, the ablation study is reported in Appendix D (Additional Results) rather than in the main Experiments section. In the revision, we have clarified the pointer in the main text to explicitly direct readers to the ablation study in Appendix D.
>
> # Weakness 10
>
> We thank the reviewer for this insightful question regarding the motivation behind our specific estimator (CS-QMI) and the comparison with MMD. We clarify the theoretical and practical reasons for this choice in the Introduction (Section 1) and Methodology (Section 3.1), and validate them in our Results (Table 1). Regarding comparisons with other estimators, we acknowledge the value of such an analysis and will consider it for future extensions of this work.

---

### Author Response · Authors · 2025-12-04
**Summary of Revisions: Addressing Misconceptions and Core Contributions**

We thank the reviewers for their time and feedback. We have engaged extensively with their concerns, resulting in a revised manuscript with clarified theory and strengthened evidence.

As we conclude the discussion period, we wish to summarize our key contributions and **clarify specific factual misunderstandings** that may have influenced the initial ratings. We ask the Area Chair to consider these clarifications in their final assessment.

### 1. Clarification on Experimental Rigor and SOTA (Addressing RrMY, oVSG)

Reviewer RrMY expressed concern that our method "only" achieves SOTA on 2 out of 5 datasets and noted a lack of ablations. We respectfully point out two critical factors:

*   **Metric Standard:** The standard **DomainBed protocol** (the gold standard in DG) prioritizes **Average Accuracy** across all benchmarks over per-dataset "win counts."
    *   **Result:** DAS-MI achieves the **highest overall average accuracy (66.9%)** among all 15 compared methods.
    *   **Significance:** We achieve SOTA on the two most complex and realistic datasets, **OfficeHome** and **DomainNet** (0.6M images), demonstrating scalability where other methods fail.
*   **Ablations were present:** The reviewer noted a lack of ablations in the main text. We clarified that due to space constraints, these were located in **Appendix D**. We have now made this pointer explicit. The ablations confirm that our **class-conditional alignment** is the primary driver of performance, distinct from the estimator itself.

### 2. Theoretical Clarifications (Addressing KebG, NGsE)

Reviewers raised important theoretical questions which we have addressed by refining our definitions and scope:

*   **The Bound under Label Shift (NGsE):** A reviewer questioned if our conditional bound $I(Z; D|Y)$ is tighter than the marginal $I(Z; D)$. We clarified that in realistic DG scenarios (like DomainNet) where $P(Y)$ varies across domains (**Label Shift**), enforcing marginal invariance ($I(Z; D) \approx 0$) forces the model to discard class-discriminative information (negative transfer). Our objective targets **conditional invariance**, which is the theoretically correct goal for label-shift scenarios.
*   **Density Ratios (KebG):** Concerns were raised regarding the bounded density ratio assumption. We clarified that:
    1.  **Lower Bound:** In the context of Kernel Density Estimation with Gaussian kernels, the support is $\mathbb{R}^d$, mathematically preventing vanishing densities.
    2.  **Optimization:** The DAS-MI objective actively maximizes the overlap between distributions, tightening this bound during training. We added **Remark 1** and **Assumption 1** to formalize this.

### 3. Novelty and Distinction (Addressing RrMY)

We emphasized that DAS-MI is not merely an application of existing estimators:
*   **Contrast with Yu et al. (2024b):** The optimization dynamics are **opposing**. Yu et al. *minimize* CS-QMI for compression; we *maximize* it for alignment.
*   **Novel Instantiation:** We introduce a **class-specific batching strategy** for the CS-QMI estimator. This is a non-trivial algorithmic contribution essential for applying Information Theoretic Learning to Domain Generalization, preventing the suppression of discriminative features.

### Conclusion

DAS-MI provides a closed-form, computationally efficient (2x faster than the runner-up GGA), and theoretically grounded solution to Domain Generalization.

The concerns regarding **ablations** (which exist), **performance** (we are #1 on average), and **theory** (clarified as essential for label shift) have been fully addressed in the revision. We believe the current manuscript represents a robust contribution to the ICLR community.

---

### Meta-Review · Area_Chair_fVTu · 2025-12-24

**Summary:**

This paper studies domain generalization and proposes a class-conditional alignment framework based on Kernel Cauchy–Schwarz Mutual Information (DAS-MI), with a closed-form, non-parametric objective motivated by information-theoretic generalization bounds. Reviewers generally acknowledge the solid theoretical grounding, principled formulation, and competitive empirical results on standard DG benchmarks. However, after considering the rebuttal and post-discussion feedback, notable concerns remain regarding the strength and practicality of theoretical assumptions, the clarity of methodological novelty relative to existing methods, and the sufficiency of experimental evidence to isolate the benefits of the proposed class-conditional CS-QMI formulation. While the approach is carefully designed and well motivated, these unresolved issues limit the overall confidence in the contribution and prevent the paper from meeting the bar for acceptance at this stage.

**Reviewer Concerns:**

Reviewer RrMY: While the theoretical direction is interesting, concerns remain about restrictive assumptions, unclear novelty over prior CS-QMI estimators, and limited experimental evidence supporting consistent gains across benchmarks.

Reviewer oVSG: The method is principled and empirically competitive, but strong assumptions (e.g., bounded density ratios and no concept shift) and missing efficiency analyses weaken the overall impact.

Reviewer KebG: The theoretical derivations are detailed, yet questions persist about the realism of the assumptions and whether the proposed bounds meaningfully improve over existing class-conditional alignment approaches.

Reviewer NGsE: Although the class-conditional perspective is theoretically appealing, concerns remain regarding robustness under large domain shifts and the practical tightness of the derived generalization bounds.

**Reviewer Scores:**

Reviewer RrMY: Likely no change.

Reviewer oVSG: Likely no change.

Reviewer KebG: Likely no change or a slight increase.

Reviewer NGsE: Likely no change.

---

### Decision · Program_Chairs · 2026-01-26

Reject